# The fate of upwelled nitrate off Peru shaped by submesoscale filaments and fronts

Jaard Hauschildt[1], Soeren Thomsen[1,2], Vincent Echevin[2], Andreas Oschlies[1], Yonss S. José[1], Gerd Krahmann[1], Laura A. Bristow[3,4], and Gaute Lavik[3]

[1]GEOMAR Helmholtz Centre for Ocean Research Kiel, Kiel, Germany
[2]Laboratoire d'Océanographie et du Climat, Expérimentations et Approches Numériques (LOCEAN), Institut de Recherche pour le Développement (IRD), Institut Pierre-Simon Laplace (IPSL), Université Pierre et Marie Curie (UPMC), Paris, France
[3]Max Planck Institute for Marine Microbiology (MPI), Bremen, Germany
[4]Department of Biology / Nordcee, University of Southern Denmark (SDU), Odense, Denmark

**Correspondence:** Jaard Hauschildt (jhauschildt@geomar.de)

**Abstract.** Filaments and fronts play a crucial role for a net offshore and downward nutrient transport in Eastern Boundary Upwelling Regions (EBUS) and thereby reduce primary production. Often studies are either based on observations or model simulations but seldom both approaches are combined quantitatively to assess the importance of filaments for primary production and nutrient transport. Here we combine targeted interdisciplinary shipboard observations of a cold filament off Peru with submesoscale-permitting (1/45°) coupled physical (CROCO) and biogeochemical (PISCES) model simulations to (i) evaluate the model simulations in detail, including the timescales of biogeochemical modification of the newly upwelled water and (ii) quantify the net effect of submesoscale fronts and filaments on primary production of the Peruvian upwelling system. The observed filament contains relatively cold, fresh and nutrient-rich waters originating in the coastal upwelling. Enhanced nitrate concentrations and offshore velocities of up to 0.5 m s$^{-1}$ within the filament suggest an offshore transport of nutrients. **Surface chlorophyll in the filament is a factor 4 lower than at the upwelling front while surface primary production is a factor 2 higher.** The simulation exhibits filaments that are similar in horizontal and vertical scale compared to the observed filament. Nitrate concentrations and primary production within filaments in the model are comparable to observations as well, justifying further analysis of nitrate uptake and subduction using the model. Virtual Lagrangian floats were released in the subsurface waters along the shelf and biogeochemical variables tracked along the trajectories of floats upwelled near the coast. In the submesoscale-permitting (1/45°) simulation 43.0 % of upwelled floats and 40.6 % of upwelled nitrate is subducted within 20 days after upwelling, which corresponds to an increase of nitrate subduction compared to a mesoscale-resolving (1/9°) simulation by 13.9 %. **Taking model biases into account we give a best estimate for subduction of upwelled nitrate off Peru between 30.5 - 40.6 %. Our results suggest** that submesoscale processes further reduce primary production by amplifying the downward and offshore export of nutrients found in previous mesoscale studies, which are thus likely to underestimate the reduction in primary production due to eddy-fluxes. Moreover, this downward and offshore transport could also enhance the export of fresh organic matter below the photic zone and thereby potentially stimulate microbial activity in the upper offshore oxygen minimum zone.

# 1 Introduction

The eastern margins of the subtropical oceans are characterized by upwelling of cold and nutrient-rich subsurface waters, caused by persistent along-shore winds that drive an offshore Ekman transport. The nutrients supplied to the sunlit surface ocean subsequently fuel high phytoplankton growth which supports a rich ecosystem (Pennington et al., 2006). These Eastern Boundary Upwelling Systems (EBUS) are found in all major ocean basins and named after the Canary, Benguela, California, and Peru-Chile current systems. The Peru-Chile upwelling system (PCUS) is the most productive EBUS in the global ocean, accounting for 10 % of the global fish catch while occupying only 0.1 % of the ocean surface (Chavez et al., 2008). The Peru upwelling ecosystem and the fisheries that depend on it thus have immense economical importance for the local population. **Furthermore, the high productivity and export of organic matter and its subsequent remineralization at depth result in high oxygen consumption (Kalvelage et al., 2015; Loginova et al., 2019). In combination with poor ventilation by sluggish currents this leads to the presence of the shallowest and most intense oxygen minimum zone (OMZ) in the global ocean (Wyrtki, 1962; Paulmier et al., 2006; Karstensen et al., 2008; Stramma et al., 2010).** Global relevance is given to EBUS by their role as natural sources of greenhouse gases to the atmosphere such as $N_2O$ (Friederich et al., 2008; Arévalo-Martínez et al., 2015) and $CO_2$ (Chavez et al., 2007; Gruber, 2015; Köhn et al., 2017; Brady et al., 2019).

The circulation in the Peru-Chile current system is characterized by opposing surface and subsurface along-shore currents: the Peru Coastal Current flows equatorward near the surface (Penven et al., 2005). The subsurface Peru-Chile Undercurrent (PCUC) flows poleward with a velocity of 10 - 15 cm s$^{-1}$ at 100 m - 150 m depth along the shelf (Wyrtki, 1963, 1967; Brink, 1983; Huyer et al., 1991; Strub et al., 1998; Chaigneau et al., 2013), supplying the nutrient-rich source waters of the coastal upwelling. Between 11 °S - 16 °S, southward velocities of 10 - 15 cm s$^{-1}$ are observed at only 25 m depth, related to a near-surfacing of the PCUC (Chaigneau et al., 2013; Pietri et al., 2014).

Mesoscale eddies have in the past been assumed to generally enhance biological productivity in the open ocean by either exposing nutrient-rich subsurface water to the well-lit euphotic zone or by lateral advection of nutrients (Falkowski et al., 1991; McGillicuddy et al., 1998; Oschlies, 2002). Conversely, in the highly productive EBUS eddies and filaments have been shown to decrease productivity by exporting nutrients and organic matter offshore and downward below the euphotic zone (Rossi et al., 2008, 2009; Lathuilière et al., 2010; Gruber et al., 2011; Nagai et al., 2015). Such features are ubiquitous in the PCUS (Penven et al., 2005; Colas et al., 2012; Thomsen et al., 2016a, b; Pietri et al., 2013; McWilliams et al., 2009). As the upwelling front meanders and eventually becomes unstable, an ageostrophic secondary circulation develops in order to restore geostrophic balance (Thomas et al., 2008; McWilliams et al., 2009, 2015). This ageostrophic flow field can drive large vertical velocities and thus impact the physical-biogeochemical coupling by modifying vertical and lateral transports of nutrients and organic matter (Lapeyre and Klein, 2006; Lévy et al., 2012; Mahadevan, 2015). The downward fluxes can be understood as subduction of surface water along isopycnals out of the mixed-layer.

**Previous studies have attempted to quantify the fluxes of biogeochemical tracers related to eddies and filaments in EBUS using biogeochemical models of various complexity (e.g. Nagai et al., 2015 in the California EBUS; Frenger et al., 2018; Montes et al., 2014; Bettencourt et al., 2015; José et al., 2017 in the PCUS; Lovecchio et al., 2018 in the**

**Canary EBUS, Schmidt and Eggert, 2016 in the Benguela EBUS)**. However, most of these studies are purely based on models and comparison to observations has proven difficult, due to the difficulties of observing vertical velocities. Also, iron is known to play a role in limiting primary production in the PCUS (Hutchins et al., 2002; Bruland et al., 2005; Browning et al., 2018) which previous model studies have not addressed with respect to eddy-fluxes of biogeochemical tracers. Regional simulations are often mainly validated using surface chlorophyll maps derived from ocean color, which does not allow to assess the underlying physical (e.g. subduction) and biogeochemical processes (e.g. primary production, hereafter PP). **For instance, if the time scale of nitrate uptake by PP was shorter than that of subduction, mainly organic matter produced in the surface layer would be subducted. If it were longer, mainly nitrate would be subducted.** When attempting to quantify the effect of subduction on biogeochemistry using models, we therefore need to ensure that the timescales of PP and subduction are realistic. **Dedicated studies combining multi-disciplinary observations with modelling efforts at meso- and submesoscale are key to advance our understanding of complex physical-biogeochemical interactions (Oschlies et al., 2018).** Evaluating the models at these scales allows to gain trust in the simulation of submesoscale processes and assess the associated uncertainties and possible systematic biases.

Furthermore, the degree to which dynamical processes of a certain scale are represented in a simulation depends on the effective horizontal resolution of the numerical model (Capet et al., 2008a; Soufflet et al., 2016). So far, coupled physical-biogeochemical model simulations focusing on eddy-fluxes of biogeochemical tracers (e.g. Nagai et al., 2015 for the California EBUS) were limited to a horizontal resolution of ~ 5 km in mid-latitudes, which is not sufficient to represent submesoscale dynamics as the effective resolution is much lower due to strong kinetic energy dissipation at the smallest resolved scales (Soufflet et al., 2016). Various purely physical model simulations (Capet et al., 2008b; Colas et al., 2012) and idealised biogeo-chemical simulations (Lathuilière et al., 2010) suggest that an increase in the horizontal resolution leads to further enhancement of horizontal and vertical fluxes.

In this study, we will focus on filaments and fronts which constitute the upper end of the submesoscale variability spectrum with length scales of $\mathcal{O}(1\text{-}10)$ km (McWilliams, 2016). To this aim we will simulate the PCUS dynamics using a coupled physical-biogeochemical model of 2.5 km resolution. A quantification of the net effect of filaments and submesoscale frontal processes on the offshore and downward nutrient transport and **PP** off Peru is missing so far. Therefore, we will address the following questions:

1. What is the amount of nitrate subduction and how does it impact PP?

2. What is the impact of horizontal model resolution on subduction and PP?

To address these questions, we will evaluate the model based on physical and biogeochemical observations of a cold filament. To assess the timescale of phytoplankton growth in our model, we will compare PP and nutrient concentrations in a modelled filament with observational data. Then, we will quantify how much of the upwelled nitrate off Peru is subducted below the euphotic zone and not taken up by biology.

This paper is structured as follows: **In section 2 the filament survey, the coupled physical-biogeochemical model and all other data sources as well as analysis methods are described. In section 3.1 the model performance with respect**

to the relevant physical and biogeochemical quantities and their horizontal variability is evaluated. In section 3.2 and section 3.3 the mean horizontal variability of the upwelling structure and cold filaments are characterized in detail both in observations and model simulations. In section 3.4 the simulation is used to analyse pathways and timescales of nitrate export, subduction and uptake and compare them to estimates from observations. In section 3.5 the effect of submesoscale-permitting vs. mesoscale model resolution on the mean biogeochemical fields is analysed. Finally, the results are discussed in the context of existing literature in section 4, which also includes a detailed discussion of the limitations of our approach. Concluding remarks follow in section 5.

## 2 Data and methods

### 2.1 Filament survey

A survey designed to investigate the biophysical coupling at a cold filament near 14 °S off the coast of Peru was carried out on April 12-17, 2017 using an adaptive sampling strategy guided by real-time satellite images. The field work was conducted during *R/V Meteor* cruise M136 which started on April 11 and ended on April 29, 2017 in Callao, Peru (Dengler and Sommer, 2017; Lüdke et al., 2019). The measurements were carried out as part of the "SFB754 - Climate-Biogeochemistry Interactions in the Tropical Ocean" project. The cruise track during the survey consisted of five transects (Fig. 1a). The first transect (CROSS) mapped the upwelling region in cross-shore direction with conductivity, temperature and depth (CTD) measurements including biogeochemical parameters ($O_2$ , $NO_3^-$, $NO_2^-$, $NH_4^+$) determined from water samples. On subsequent along-shore transects, a cold filament present ~ 100 km southeast of transect CROSS was crossed by *R/V Meteor* four times in a zigzag pattern: Twice with high-resolution physical underway CTD measurements heading in southeast direction (PHY, PHY2) and two more times with station-based lowered CTD measurements including biogeochemical parameters heading in northwest direction (BIO, BIO2). A dense sampling strategy with 8 - 10 km horizontal spacing between stations and 5 - 10 m vertical spacing between samples was applied for the biogeochemical transects. The physical underway transects (PHY, PHY2) were completed overnight in under 8 hours sampling with a horizontal spacing of under 1 km, similar to the resolution of the binned ADCP data. This physical data thus closely represents a synoptic view of the surface ocean. Sampling on the biogeochemical transects (BIO, BIO2) was done during daytime following each physical transect. Wind speed and direction on *R/V Meteor* were measured at 35.5 m height with a temporal resolution of one minute and corrected to 10 m height following Smith (1988), similar to the procedure used by Köhn et al. (2017).

### 2.2 Oceanographic biophysical measurements

Hydrographic data was obtained from lowered conductivity, temperature and pressure (CTD) measurements using SeaBird SBE 9-plus CTD system equipped with two sets of pumped sensors. Water samples for oxygen, nutrients and salinity were taken using 24 Niskin bottles (10 l) mounted on a General Oceanics rosette. Salinity samples were analyzed on board with a Guildline Autosal 8 model 8400B salinometer to calibrate conductivity measurements to practical salinity (PSS-78) with an uncertainty

of 0.003 g kg$^{-1}$. Practical salinity was converted to absolute salinity (TEOS-10) using routines of the Gibbs Seawater toolbox (https://github.com/TEOS-10/GSW-Python). The CTD was also equipped with an oxygen sensor and a WET Labs (USA) fluorometer. The oxygen sensor was calibrated to an accuracy of 1.5 $\mu$mol using Winkler titration. As Winkler titration is not reliable in the core of the OMZ and results in too high values (Revsbech et al., 2009; Kalvelage et al., 2013; Thomsen et al.,

2016b), a concentration of 0 $\mu$mol l$^{-1}$ was assumed in the core of the OMZ and the profiles corrected accordingly following Langdon (2010). To determine chlorophyll-$a$ concentrations from the measured chlorophyll fluorescence, the original factory calibration provided by the sensor manufacturer WET Labs (USA) was used. For more details on calibration of chlorophyll fluorescence measurements, the reader is referred to Loginova et al. (2016). Underway subsurface temperature and salinity were measured using a Teledyne Oceanscience (Poway, USA) RapidCAST system acquiring profiles of the upper 70 m of the water

column every 2 minutes resulting in a horizontal resolution of $790 \pm 240$ m depending on the vessel speed. Subsurface current velocities on *R/V Meteor* were recorded by a vessel-mounted Acoustic Doppler Current Profiler (vmADCP). The system used was a Teledyne RD Instruments OceanSurveyor 75kHz ADCP capable of reaching a maximum depth of ~ 700m. The shallowest velocity measurements were acquired in a bin centered 18 m below the sea-surface. During the filament crossing, the vessel speed was kept nearly constant at ~ 5 m s$^{-1}$ to obtain high-quality velocity measurements with a vertical resolution

of 8 m and a horizontal resolution of $290 \pm 26$ m which was subsequently averaged in 1 km bins. Nutrient concentrations were determined onboard by a QuAAtro autoanalyzer (SEAL Analytical, Southampton, UK) using standard photometric methods (Grasshoff et al., 1983).

During M136 a self-contained ultraviolet SUNA nitrate sensor manufactured by Sea-Bird Scientific was attached to the CTD/Rosette system similar to Alkire et al. (2010). SUNA sensors determine the concentration of nitrate by measuring the

absorption of UV light over a fixed path length. The SUNA data has been reprocessed with the concurrent CTD pressure, temperature, and salinity (for bromide absorption) data to eliminate their effects on the absorption and the resulting nitrate concentrations (Sakamoto et al., 2009, 2017). The resulting SUNA nitrate concentrations have been extracted for the times at which bottles were closed on the watersampler. These concentrations have then been compared to the nitrate and nitrite concentrations measured with the Autoanalyzer. SUNA nitrate values correlated highly with the Autoanalyzer values (r-squared

0.9972, with the 10% most deviating samples removed). SUNA values were also compared to the combined Autoanalyzer concentrations of nitrate and nitrite but the resulting correlation was somewhat weaker (r-squared 0.9964, with the 10% most deviating samples removed). The SUNA measurements were thus corrected to match the nitrate (NO$_3$) concentration by applying the following correction term where NO$_{3, \text{old}}$ is the original measurement and NO$_{3, \text{new}}$ is the final corrected value: NO$_{3, \text{new}}$ = 1.2813 + 1.0576 × NO$_{3, \text{old}}$. We applied this calibration to all SUNA nitrate concentrations.

## 2.3   Incubations

Seawater samples were filled into 2 L polycarbonate bottles, and were stored in the dark until tracer additions were made, which was always within 2 hours of collection. Following the method outlined in Großkopf et al. (2012), incubations were started with the addition of sodium bicarbonate (NaH$_{13}$CO$_3$; > 98 atom%, Sigma Aldrich) to yield an enrichment of approximately

3.2 atom% final. At each depth sampled, three bottles received a $^{13}$C addition, and a fourth bottle received no $^{13}$C and acted as an untreated control allowing the natural abundance $^{13}$C to be determined at each depth. All bottles were placed into on-deck incubators with surface seawater flow-through and shaded with 20, 10 or 1% surface irradiance (Lee Filters, Seattle, WA, USA), depending on the sampling depth. Incubations were terminated after 24 hours by filtration onto 25mm pre-combusted (450°C, 4 h) GF/F filters (Whatman), which were dried onboard (50°C, 12 h) and stored at room temperature until analysis. Prior to analysis GF/F filters were acidified over fuming HCl overnight in a dessicator, dried and pelletized in tin cups. Samples were analyzed for particulate organic carbon and isotopic composition using continuous flow isotope ratio mass spectrometry coupled to an elemental analyzer. PP rates were calculated from the incorporation of $^{13}$C into biomass as described in Großkopf et al. (2012).

## 2.4   Data products

To guide the shipboard measurements and put them into a regional context MODIS (Moderate Resolution Imaging Spectroradiometer) Level 2 along-track sea-surface temperature (SST) and chlorophyll-*a* products with an approximate resolution of 1 km from the TERRA and AQUA satellites were used (https://oceandata.sci.gsfc.nasa.gov/). We restricted our analysis to daylight images of SST and used the cloud mask based on ocean color because of obvious deficiencies of the cloud mask based on infrared SST data alone. SST data from AVHRR (Advanced Very High Resolution Radiometer) at 25 km resolution was downloaded from NOAA (ftp://eclipse.ncdc.noaa.gov/pub/OI-daily-v2/NetCDF/). **For evaluating the model performance with respect to PP, we used estimates of ocean Net Primary Production (NPP) which were derived from both MODIS and SeaWIFS chlorophyll-*a* using the Vertically Generalized Production Model (VGPM, Behrenfeld and Falkowski, 1997, http://sites.science.oregonstate.edu/ocean.productivity/). A global mixed-layer depth (MLD) climatology with 2° x 2° resolution based on a 0.2°C temperature criterion was provided by IFREMER (http://www.ifremer.fr/cerweb/deboyer/mld/Surface_Mixed_Layer_Depth.php). Annual mean temperature and nitrate fields for model evaluation were provided at 1/2° resolution by the CARS climatology (CSIRO Atlas of Regional Seas, Ridgway et al., 2002, http://www.marine.csiro.au/atlas/). Annual mean gridded chlorophyll-*a* products from the MODIS and SeaWIFS satellite-based instruments were downloaded from NOAA (https://oceandata.sci.gsfc.nasa.gov/).**

## 2.5   Coupled Physical-Biogeochemical Model (CROCO/PISCES)

We employed CROCO (Coastal and Regional Ocean Community model) to study the circulation in the PCUS at submesoscale-permitting resolution. CROCO is a free-surface, terrain-following coordinate ocean modeling system built upon ROMS_AGRIF (Penven et al., 2006; Shchepetkin and McWilliams, 2009) and a non-hydrostatic kernel (not used in this study). CROCO solves the Primitive Equations using the Boussinesq approximation and a hydrostatic vertical momentum balance. **The nonlocal K-Profile Parameterization (KPP) scheme is used to handle unresolved processes related to vertical mixing.** For a complete description of the model numerical schemes the reader can refer to Shchepetkin and Mcwilliams (2005). The code used in the present study is the CROCO v1.0 version, which is very similar to ROMS_AGRIF version v3.1.

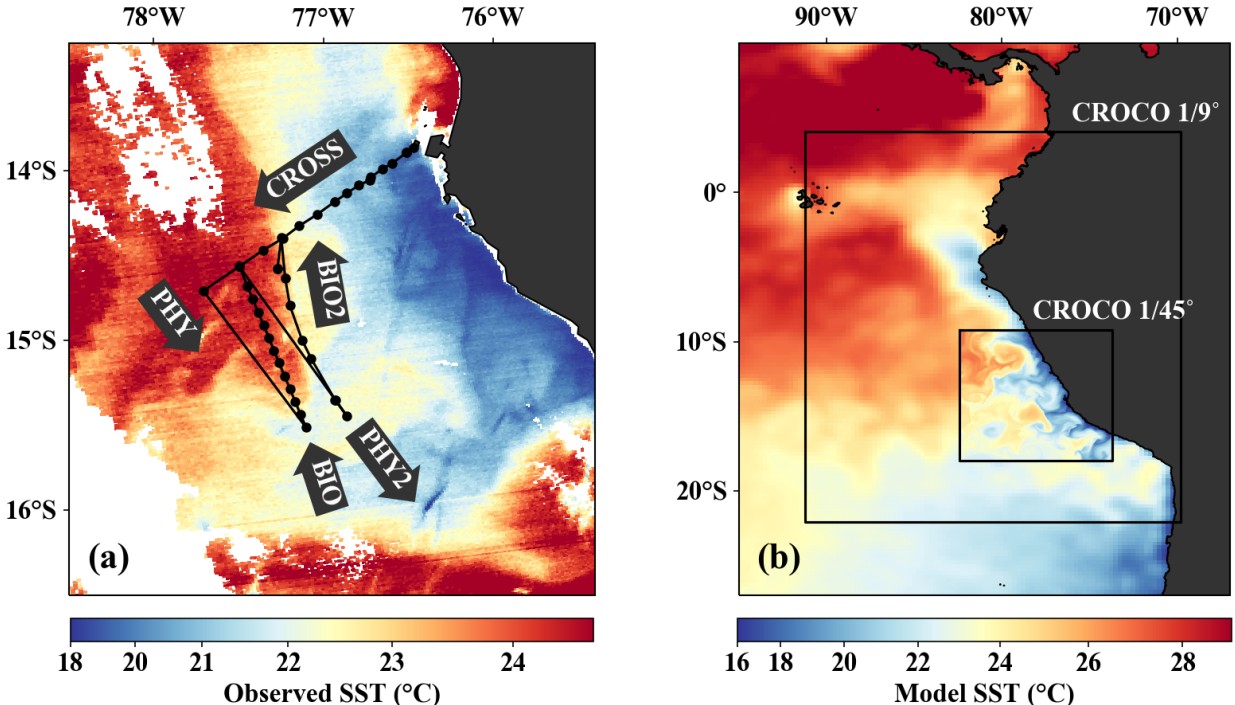

**Figure 1.** (a) Observed SST (MODIS) on April 14, 2017 with cruise track and section names superimposed (b) Model SST on April 14, 2017 in the coarse- (1/9 °) and high-resolution (1/45 °) CROCO simulations superimposed on AVHRR satellite SST. Black rectangles indicate the respective model domains.

The model was configured as a nested set of two spatial domains (Fig. 1b) using an offline, one-way embedding procedure (Mason et al., 2010). The outer domain has a resolution 1/9 ° over a region of 2207 km in zonal direction by 2911 km in meridional direction (24.4 ° x 26.2 °), and the inner domain has a resolution of 1/45 ° over a region of 918 km in zonal direction by 973 km in meridional direction (8.69 ° x 8.76 °). **Since the Peruvian upwelling system is located relatively close to the**
5 **equator, the Rossby radius is about 70 km in our study area (Chelton et al., 1998), an order of magnitude larger than our model resolution (~ 2.5 km). The Rossby radius represents effectively the limit of mesoscale dynamics and we can thus consider our model submesoscale-permitting, as it resolves the upper range of the submesoscale variability spectrum.** There are 32 sigma-levels and the vertical resolution varies with water depth. Here we focus on the upper 200 m of the water column where the vertical resolution near the surface is 1 - 2 m / 5 m on the shelf / offshore. The model topography was derived
10 from the GEBCO (General Bathymetric Chart of the Oceans, http://www.gebco.net) product, interpolated onto the model grid and smoothed to reduce pressure gradient errors.

The lateral open boundary conditions of the outer domain for temperature, salinity, velocities and sea-level were provided at 1/12 ° resolution by the MERCATOR PSY4V2 model (Lellouche et al., 2018) which assimilated in-situ data transmitted from *R/V Meteor* during the research cruises M135 (large scale mapping off the OMZ, https://doi.pangaea.de/10.1594/PANGAEA.

890441) and M136 (https://doi.pangaea.de/10.1594/PANGAEA.892564) and from ARGO floats (http://www.argo.ucsd.edu/), as well as satellite SST and sea-level measurements. No assimilation or "nudging" was done inside the model domain except for a restoring term on the surface heat flux. The net surface heat flux $Q$ is given by the COADS (Worley et al., 2005) climatology relaxed to AVHRR (Advanced Very High Resolution Radiometer; ftp://eclipse.ncdc.noaa.gov/pub/OI-daily-v2/NetCDF/) daily
SST according to

$$Q = Q_{\text{COADS}} - \frac{dQ}{dT} \cdot (\text{SST}_{\text{CROCO}} - \text{SST}_{\text{AVHRR}}) \tag{1}$$

where $\frac{dQ}{dT}$ represents the additional heat flux that is imposed per degree of temperature difference between model SST and observed SST. This heat flux correction is a function of atmospheric parameters and assumes values of $30\text{-}35$ W/m$^2$/$^\circ$C (Barnier et al., 1995). The model was forced with surface wind stress derived from the daily level-2 wind product provided by
the ASCAT scatterometer (https://podaac.jpl.nasa.gov/dataset/ASCATB-L2-25km).

The CROCO model was coupled to the PISCES (Pelagic Interaction Scheme for Carbon and Ecosystem Studies) biogeochemical model which simulates the biogeochemical cycles of carbon and the main nutrients (P, N, Si, Fe). It includes two phytoplankton compartments (nanophytoplankton and diatoms), two zooplankton size classes (microzooplankton and mesozooplankton), two detritus classes and a description of the carbonate chemistry. **A detailed model description is given in**
**Aumont et al. (2015), in the following we will outline only the equations that are relevant for our analysis of the local temporal nitrate changes. The evolution of nitrate in PISCES is determined by equation (2), with the right-hand side terms representing nitrate increase due to nitrification and nitrate loss due to small phytoplankton growth, large phytoplankton growth and denitrification (we omitted the physical transport and mixing terms):**

$$\frac{\partial \text{NO}_3}{\partial t} = \text{Nitrif} - \mu_{\text{NO}_3}^P P - \mu_{\text{NO}_3}^D D - \text{Denitrif} \tag{2}$$

**The nitrate uptake rate of small phytoplankton $\mu_{\text{NO}_3}^P$ is defined by equation (3) as follows:**

$$\mu_{\text{NO}_3}^P = \mu^P \frac{L_{\text{NO}_3}^P}{L_{\text{NO}_3}^P + L_{\text{NH}_4}^P} \tag{3}$$

**The limitation term for nitrate $L_{\text{NO}_3}^P$ is described by equation (4):**

$$L_{\text{NO}_3}^P = \frac{K_{\text{NH}_4}^P \text{NO}_3}{K_{\text{NO}_3}^P K_{\text{NH}_4}^P + K_{\text{NH}_4}^P \text{NO}_3 + K_{\text{NO}_3}^P \text{NH}_4} \tag{4}$$

**The limitation term for ammonium $L_{\text{NH}_4}^P$ is similar to equation (4) but with the product of ammonium concentration**
**$\text{NH}_4$ and half-saturation constant $K_{\text{NO}_3}^P$ in the numerator. The half-saturation constants for nitrate $K_{\text{NO}_3}^P$ and for ammonium $K_{\text{NH}_4}^P$ set the concentrations at which the limiting effect of each nutrient would result in half the maximum growth rate. The growth rate for small Phytoplankton $\mu^P$ is described by equation (5),**

$$\mu^P = \mu_{\max}^0 f_1(T) f_2(L_{\text{day}}, z_{\text{mxl}}) \left( 1 - \exp\left( \frac{-\alpha^P \theta^{\text{Chl},P} \text{PAR}^P}{L_{\text{day}} \mu_{\max}^0 f_1(T) L_{\text{lim}}^P} \right) \right) L_{\text{lim}}^P \tag{5}$$

where $\mu_{\mathbf{max}}^{0}$ is the maximum growth rate at $0°\mathrm{C}$ and $f_1$ is a function describing the dependence on the temperature $T$ of the growth rate. The function $f_2$ introduces additional dependencies of the growth rate on the length of day $L_{\mathbf{day}}$ and the mixed layer depth $z_{\mathbf{mxl}}$ in case it exceeds the euphotic zone. The term inside the parentheses is defined so that it increases exponentially with the amount of absorbed light given by the product of a constant parameter $\alpha^P$, the variable chlorophyll to carbon ratio $\theta^{\mathbf{Chl},P}$ and the photosynthetically available radiation $\mathbf{PAR}^P$. The total nutrient limitation term $L_{\mathbf{lim}}^P$ is defined in such a way (Equation 6) that the nutrient with the smallest individual limitation term (i.e. the smallest concentration relative to its half-saturation constant) is taken to be the limiting nutrient and controls phytoplankton growth. Since phytoplankton can use both nitrate and ammonium as a nitrogen source, the individual limitation terms for these nutrients are added before calculating the total nutrient limitation term.

$$L_{\mathrm{lim}}^{P} = \min\left( L_{\mathrm{PO}_4}^{P}, L_{\mathrm{NO}_3}^{P} + L_{\mathrm{NH}_4}^{P}, L_{\mathrm{Fe}}^{P} \right) \tag{6}$$

The equations for the nitrate uptake rate of large phytoplankton $\mu_{\mathrm{NO}_3}^{D}$ (see equation 2) have an identical structure to equations 3-6 and are therefore omitted here. Primary production is proportional to the sum of the small phytoplankton biomass $P$ and large phytoplankton biomass $D$, multiplied by their respective growth rates $\mu^P$ and $\mu^D$.

The open boundary conditions for oxygen, nitrate, phosphate and silicate were provided at $1/2°$ resolution by the CARS climatology (CSIRO Atlas of Regional Seas, Ridgway et al., 2002) as the sum of an annual mean and both annual and semi-annual cycles. For the variables not available from data climatologies (iron, dissolved organic & inorganic carbon, total alkalinity), a climatology derived from model output of a global NEMO-PISCES simulation at $2°$ resolution was used (Aumont et al., 2015). After a spinup of 14 years for the $1/9°$ simulation, the $1/9°$ and $1/45°$ simulations were started from this model state and run from January 2013 until April 2017. Only model data from March 2014 onwards has been used in the analysis to allow for an additional spinup of the submesoscale dynamics in the $1/45°$ nest. **The model output frequency was set to 1 day for the $1/45°$ simulation and 3 days for the $1/9°$ simulation, except for the Lagrangian analysis where we generated additional output at 4-hour frequency from both simulations to allow for a more precise offline calculation of the float trajectories and ensure that the results are comparable.**

## 2.6 Virtual Lagrangian float experiment

To study the temporal evolution of biogeochemical properties in the upwelled water, an ensemble of 20 float experiments was conducted. The ensemble consist of 5 experiments each performed in April of the years 2014 - 2017 and initialised on day 1, 6, 11, 16 and 21. For each of these experiments, 250.000 virtual Lagrangian floats were advected by the 4 h average model flow field for 35 days using the ROMS offline tool (Capet et al., 2004; Carr et al., 2008). Virtual floats were released between the coast and ~250 km offshore over the upper 150 m and biogeochemical variables were tracked along float trajectories. Following Thomsen et al. (2016a) we subsampled the trajectories of all floats that (1) are in the euphotic zone at a given time, (2) were below the euphotic zone for at least 1 day before that and at the time of release, (3) have a density higher than $25\,\mathrm{kg\,m}^{-3}$ and (4) are located between $13°\mathrm{S}$ and $16°\mathrm{S}$ which we then consider as upwelled. The euphotic zone is defined as

the upper layer of the ocean where photosynthetically active radiation is > 1% of its surface value. The duration of 1 day was chosen somewhat arbitrarily to exclude floats that have their source at the surface and are simply subject to relatively short, alternating vertical motions while they enter the upwelling patch. This is justified as the upwelling implies a source at the subsurface and the results are not sensitive to this parameter choice. The density criterion (3) is imposed to restrict our analysis to the trajectories that surface inshore of the upwelling front, where the densest isopycnals outcrop. The regional criterion (4) ensures that the upwelled floats originate close to the center of the model domain and will only rarely reach the open boundaries during the experiment. It also serves the purpose of maintaining comparability with the observational data that was collected in this region.

To analyse the fate of upwelled nitrate in more detail, we used the subsampled float trajectories to compute the fraction of upwelled nitrate that is subducted. We computed a "subduction ratio" as follows:

$$\text{ratio} = \frac{\sum_{i=1}^{N_{\text{subducted}}} \text{NO3}_{i,t_{20}}}{\sum_{i=1}^{N_{\text{upwelled}}} \text{NO3}_{i,t_0}} \tag{7}$$

where $N_{\text{upwelled}}$ is the total number of upwelled floats, $N_{\text{subducted}}$ is the total number of subducted floats, $\text{NO3}_{t_0}$ is the nitrate concentration of each float at the time of upwelling and $\text{NO3}_{t_{20}}$ is nitrate concentration of each float 20 days after upwelling. We first save the nitrate concentration for each individual float at the time of upwelling. If a float is below the photic zone - defined as 0.1% surface intensity PAR - 20 days after upwelling, we consider this float subducted and also save the nitrate concentration at this time. The sum of the nitrate concentration over the subducted floats divided by the sum of the nitrate concentration over all upwelled floats yields the nitrate subduction ratio. Using this ratio we can account for the reduction in nitrate during the time period that the subducted floats stay in the photic zone. A timescale of 20 days was chosen because the number of upwelled floats below the photic zone appears to **have stabilized** after this time (Fig. 6f).

## 3 Results

### 3.1 Model evaluation

**Before using the submesoscale-permitting simulation to analyse the process of nitrate subduction in the Peruvian up-welling, we verify that the simulation realistically captures the relevant physical and biogeochemical dynamics. The model evaluation focuses here on time-averaged mean quantities and their horizontal variability as a detailed comparison of the simulation to observations on synoptic scales follows in the next subsections. The 2-year averaging period (2015-2016) is chosen to allow for sufficient spin-up of small-scale dynamics in the $1/45°$ simulation, which was initialised on January 1, 2013.**

**The model fit of the most relevant physical and biogeochemical variables to observations is summarised in Taylor diagrams for both the $1/45°$ and $1/9°$ simulations (Fig. 2a,b). The simulated SST shows only a small negative bias ($\sim -3\%$) relative to satellite observations for both simulations. The spatial patterns of simulated and observed mean SST are**

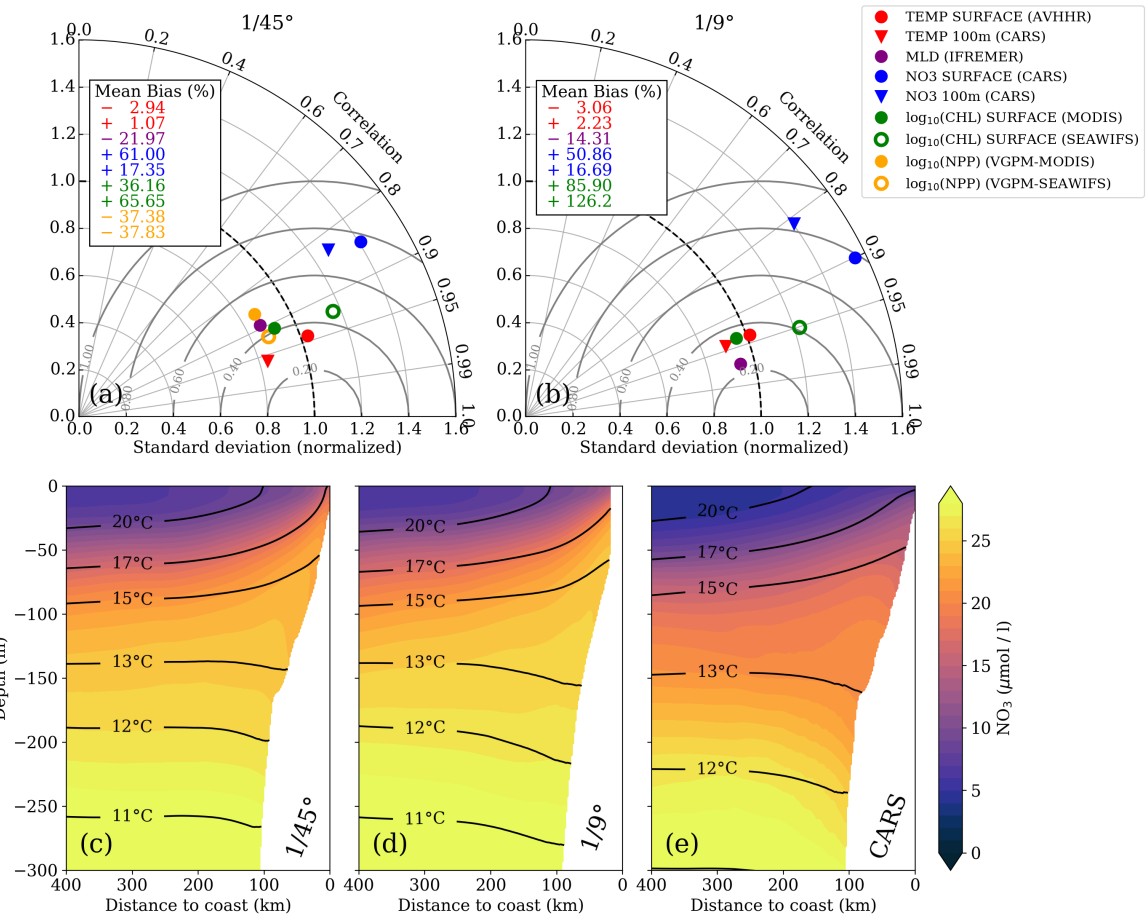

**Figure 2.** Taylor diagrams for (a) 1/45° and (b) 1/9° simulations, respectively. The statistics were computed from horizontal mean fields (see Fig. S1) and therefore represent their spatial variability. Due to the strongly skewed distributions of chlorophyll-$a$ and PP these variables were logarithmized prior to computing correlation and standard deviation to approximate normally distributed variables, while the mean bias was calculated using the original values. Grey concentric circles centred around unity normalised standard deviation and correlation indicate the normalised root mean square error (NRMSE). The analysis for the 1/9° simulation was restricted to the extent of the 1/45° domain to facilitate direct comparison. Primary production rates were not saved for the 1/9° simulation and could therefore not be evaluated. (c-e) Along-shore averaged sections of mean nitrate concentrations with temperature contours superimposed for (c) 1/45° simulation, (d) 1/9° simulation and (e) CARS climatology. The CARS climatology was interpolated onto the 1/45° model grid before computing the along-shore average.

also highly correlated (r=0.95) and their standard deviations are almost identical, resulting in a normalized standard deviation of about unity (NSTD≈1). This is expected since the same SST observations are also used to calculate the restoring term on the surface heat flux at runtime of the model, effectively constraining the model SST. At 100 m depth, the temperature bias relative to the CARS climatology is smaller and of the opposite sign compared to the surface for

both the 1/9° simulation (+2.23%) and the 1/45° (+1.07%) simulation. The correlation between model and observations at this depth remains high and the modelled standard deviation also compares well with the climatological value (r≈0.95, NSTD≈0.8-0.9). The normalised root mean square error for the simulated temperature fields at the surface and at 100 m depth is therefore low (NRMSE≈0.35). The mixed-layer depth (MLD) has a negative bias (−14.31%) with respect to the IFREMER climatology in the 1/9° simulation, while the spatial variability of the mean shows a very good

fit (r>0.95, NRMSE≈0.25). In the 1/45° simulation the MLD bias is slightly larger (~ −22%) and also with respect to the spatial variability the fit is slightly worse (r≈0.9, NRMSE≈0.45). Note that the gridded MLD climatology has a coarse resolution (2° x 2°), which may contribute to the weaker correlation with the MLD in the 1/45° simulation. Chlorophyll-$a$ shows quite a substantial positive bias in both simulations relative to MODIS (>30%) as well as SeaWIFS (>60%) satellite observations (Fig. 2a,b). This positive bias is reduced in the 1/45° simulation by a factor of 2 compared

to the 1/9° simulation. Despite the mean bias, the model fit is quite good with respect to the spatial chlorophyll variability relative to both MODIS and SeaWIFS data (NRMSE≈0.35-0.45). Modelled PP in the 1/45° simulation exhibits a negative bias relative to PP estimates derived from ocean color (~ −37%). The spatial correlation of this modelled PP to the satellite estimate is high, and the standard deviation is only slightly underestimated in the simulation (r≈0.9, NSTD≈0.8), resulting in a good model fit (NRMSE≈0.45, Fig. 2a).

Nitrate in both simulations shows a substantial bias at the surface (~ +51-61%) and a much smaller bias at 100 m depth (~ +17%), which is also apparent in figure 2c-e. There is however a high spatial correlation of the modelled mean nitrate fields with climatological fields (r≈0.8-0.9) and the standard deviation in the simulations slightly overestimated compared to the climatology (NSTD≈1.2-1.5). In general the model fit with respect to nitrate is improved slightly in the 1/45° simulation (NRMSE≈0.75, Fig. 2a) compared to the 1/9° simulation (NRMSE≈0.8, Fig. 2b).

To evaluate how the simulated mean horizontal and vertical temperature and nitrate gradients compare, we computed along-shore averaged mean sections of these variables from both simulations as well as from the CARS climatology (Fig. 2c-e). Above 100 m depth, which corresponds to about 13°C, the location of the simulated isotherms compares well with the climatology, but the model isotherms are slightly steeper within 200 km from the coast and less steep further offshore. Below 100 m depth, the vertical temperature gradient in the simulations is about 0.5°C m$^{-1}$ higher

than in the climatology.

The vertical nitrate gradient below 100 m depth is weaker in the simulations compared to the CARS climatology (Fig. 2c-e). The horizontal nitrate gradient reverses at ~ 150 m depth, a feature which is present in the modelled and the climatological nitrate fields. Despite the aforementioned bias of the simulated mean nitrate distribution, the horizontal and vertical nitrate gradients above 100 m depth compare well with the CARS climatology (Fig. 2c-e). We can therefore

assume that the model can realistically represent the nitrate fluxes associated with upwelling and subduction, which

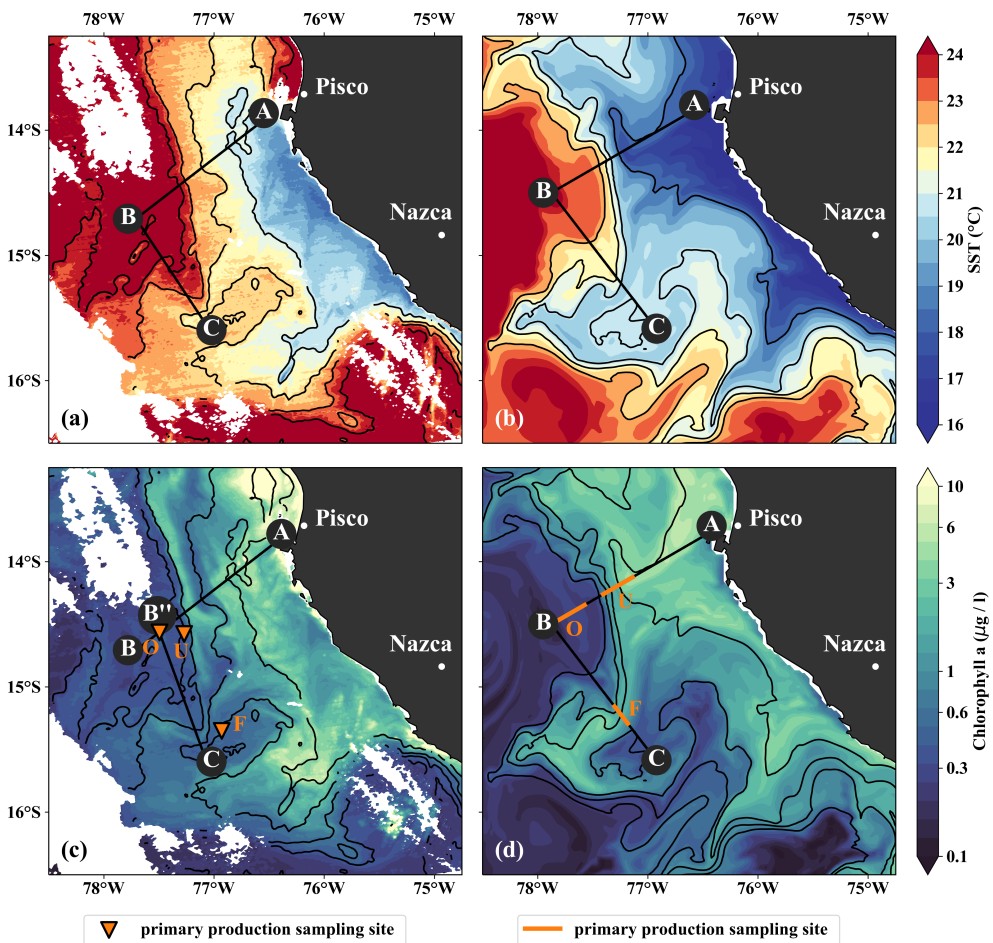

**Figure 3.** (a,b) Sea-surface temperature and (c,d) Surface chlorophyll in (a,c) observations on April 14, 2017 and in (b,d) the model simulation on April 5, 2017. Locations of vertical sections are superimposed (see Figs. 4; 5). Orange triangles in (c) indicate three sampling locations offshore (O), at the upwelling front (U) and in the filament (F) where primary production was measured. Orange lines in (d) indicate the corresponding locations used for comparison in the simulation. **The simulated fields represent 1-day averaged model output, which was chosen for the the horizontal gradients to be as sharp as possible and directly comparable to observations.**

occur predominantly in this depth range. For a more detailed evaluation of the 1/9° simulation please refer to Echevin et al. (2008), where a similar model configuration was used. The same simulations as in this study were used in a Master thesis by Hauschildt (2017), where an in depth model evaluation can be found.

## 3.2 Physical and biogeochemical upwelling structure in observations and simulations

The filament survey (Sec. 2.1) was carried out during the transition from Austral summer to fall on April 12 - 17, 2017. Being typical for the season, moderate southeasterly along-shore winds between 5 - 6 m s$^{-1}$ near the coast and 11 - 14 m s$^{-1}$ offshore were observed throughout the survey, which represents upwelling-favourable conditions (not shown). The most intense upwelling is often found in distinct cells near headlands and capes, indicated by along-shore minima of sea-surface temperature (SST). A well-known upwelling cell off Peru can be identified near 15 °S by its relatively low SST (18 °C) in a satellite image taken on April 14, 2017, 18:25 UTC (Fig. 3a). A strong cross-shore SST gradient exists between this coastal minimum and the warm offshore waters (24.5 °C). The maximum SST gradient (0.15 °C/km) is found 110 - 130 km offshore along the 23 °C isotherm and marks the location of the upwelling front. The offshore increase in SST is accompanied by an increase in salinity from 35.3 to 36.25 g kg$^{-1}$ and an increase in mixed-layer depth from 5 m to 30 m, approximately following the $\sigma_\theta = 25$ kg m$^{-3}$ isopycnal (Fig. 4a). Offshore of the upwelling front, a sharp thermocline with vertical temperature gradients of up to 0.4 °C m$^{-1}$ across the base of the mixed layer is found (Fig. 4a). The deepest isopycnal that outcrops at the coast ($\sigma_\theta = 25.5$ kg m$^{-3}$) descends to 50 m depth 75 - 100 km offshore. The predominant water mass along the density surfaces that supply the coastal upwelling is the Equatorial Subsurface Water (ESSW, e.g. Silva et al., 2009) at a density of about $\sigma_\theta = 26.0$ kg m$^{-3}$ and a salinity of 35.2 g kg$^{-1}$ (Fig. 4b), which is transported poleward along the shelf by the Peru-Chile Undercurrent (PCUC; Gunther, 1936; Fonseca, 1989; Montes et al., 2010). The maximum poleward velocities of ~ 0.5 m s$^{-1}$ are observed within 50 km from the coast at 20 - 100 m depth (Fig. 4c).

The observed physical variability in the upwelling region gives rise to biogeochemical variability on similar scales (Figs. 3c; 4d,e). Surface chlorophyll concentrations are enhanced inshore of the upwelling front (~ 5 mg m$^{-3}$) compared to offshore (~ 0.3 mg m$^{-3}$) due to nutrient-rich subsurface waters (15 - 20 $\mu$mol l$^{-1}$ NO$_3$) that are brought to the surface in the coastal upwelling (Figs. 3c; 4d,e). Surface NO$_3$ concentrations decrease continuously by about 0.1 $\mu$mol l$^{-1}$ per kilometer cross-shore distance to 5 $\mu$mol l$^{-1}$ inshore of the upwelling front (Fig. 4e). Note that a local chlorophyll maximum occurs on the cold side of the upwelling front (~ 10 mg m$^{-3}$, Figs. 3c; 4d). Offshore of the upwelling front surface nutrients are depleted (< 1 $\mu$mol l$^{-1}$ NO$_3$) and a strong vertical gradient of up to 2 $\mu$mol l$^{-1}$ NO$_3$ m$^{-1}$ is present across the base of the mixed-layer (Fig. 4e). As a result, the maxima in chlorophyll (7 - 10 mg m$^{-3}$, Fig. 4d, Table 1) and PP (~ 9 $\mu$mol C l$^{-1}$ d$^{-1}$, Table 1) occur below the mixed-layer where nutrients are abundant (20 $\mu$mol l$^{-1}$ NO$_3$, Fig. 4e). Below 80 m depth chlorophyll concentrations are low (< 0.2 mg m$^{-3}$) everywhere in the study area (Fig. 4d). Due to similarly low subsurface chlorophyll concentrations in the source waters on the shelf, surface chlorophyll concentrations remain relatively low (~ 1 mg m$^{-3}$) within 20 - 30 km from the upwelling center and only peak (4 - 6 mg m$^{-3}$) beyond this area (Figs. 3c; 4d). This illustrates that despite the clear inverse relationship of chlorophyll and SST on larger scales, small-scale chlorophyll variability is more complex and not consistently related with SST.

The characteristic structure of coastal upwelling in the physical fields is well reproduced in our simulations, but some differences exist (Figs. 4a-c,f-h). The location of the upwelling front 100 km offshore and the corresponding $\Delta$ SST maximum of 0.2 °C km$^{-1}$ agrees well with both satellite images and in-situ measurements (Figs. 3a,b; 4a,f). The temperature and salinity

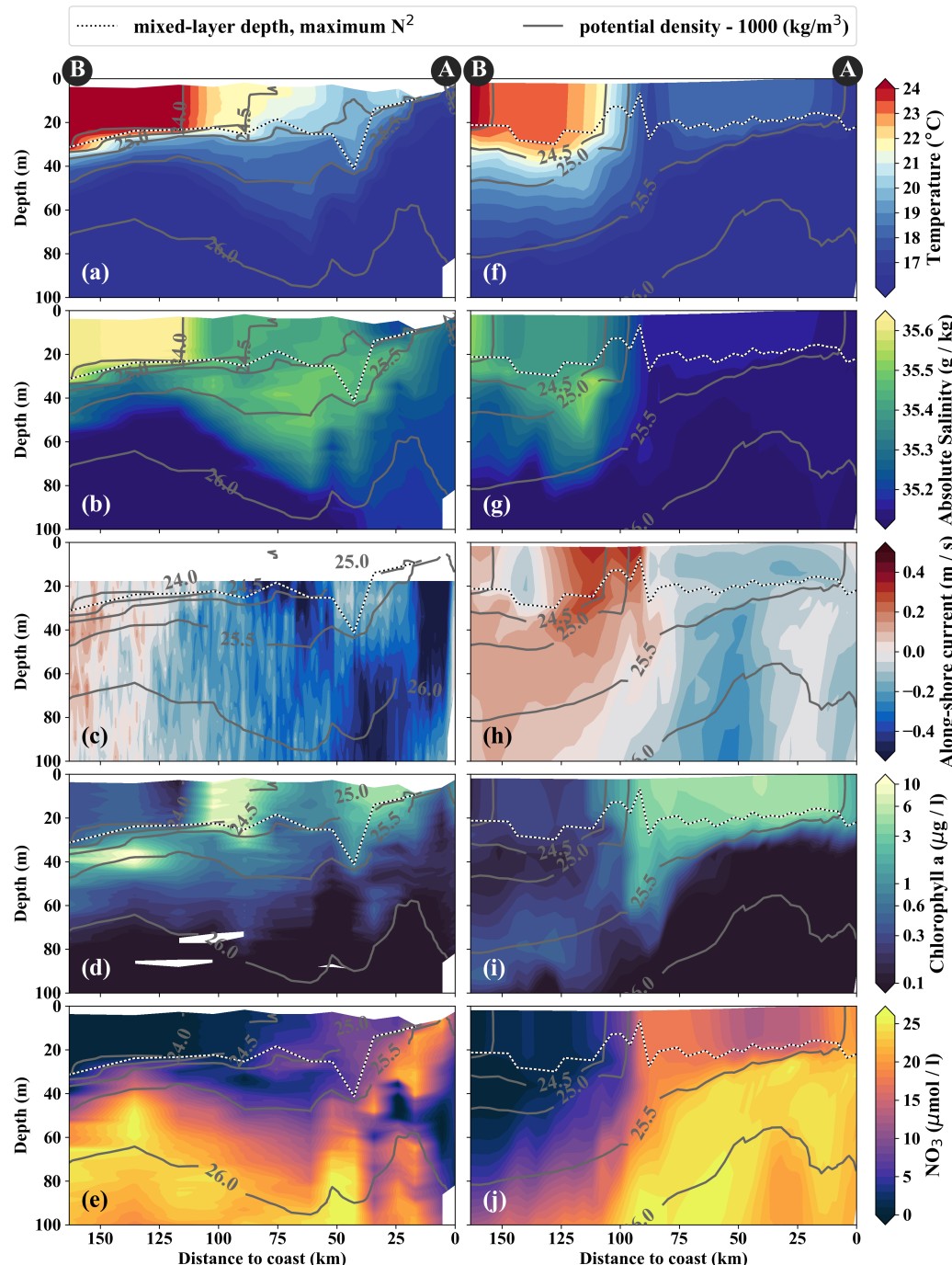

**Figure 4.** Cross-shore sections of (a,f) temperature, (b,g) salinity, (c,h) along-shore current, (d,i) chlorophyll and (e,j) nitrate in observations (a-e) and model simulation (f-j). Potential density contours are shown in grey and mixed layer depth is represented by the broken white/black line. Letters A and B indicate the section endpoints marked in Fig. 3. **The model sections represent 1-day averaged output on April 14, 2017, which was chosen for the horizontal and vertical gradients to be as sharp as possible and directly comparable to observations.**

distributions are similar overall to observations in the simulation, apart from a cold / fresh bias of the surface waters inshore of the upwelling front (Figs. 4a,b,f,g). Due to this bias the $\sigma_\theta = 25$ kg m$^{-3}$ isopycnal outcrops 100 km offshore in the simulation and near the coast in the observations. The average mixed-layer depth in the simulation is ~ 20 m, very close to observed values offshore of ~ 100 km. However, the observed mixed-layer depth decreases to only ~ 5 m in the coastal upwelling patch, whereas such a shallow mixed-layer is not seen in the simulation (Fig. 4a,f). The southward velocities of ~ 0.3 m s$^{-1}$ inshore of the upwelling front in the simulation that are associated with the surfacing undercurrent are similar to observations (Fig. 4c,h). However, the strongest southward flow (~ 0.3 m s$^{-1}$) in the simulation is weaker than observed (~ 0.5 m s$^{-1}$) and not located at the shelf but 55 km offshore. This is likely related to differences in the mesoscale variability, since an anticyclone is present immediately offshore of the upwelling patch in the simulation compared to a cyclonic eddy at approximately the same position in the observations (not shown). However, averaging over the period 2015 - 2016 yields an alongshore velocity of 13 cm s$^{-1}$ in the core of the PCUC, which is close to the observed velocity (~ 14 cm s$^{-1}$, Chaigneau et al., 2013).

In the simulation, the upwelling structure also dominates the variability of the biogeochemical fields (Figs. 3d; 4i,j). Chlorophyll concentrations larger than 0.2 mg m$^{-3}$ are found down to 80 m (100 m) in the observations (simulation), showing overall good agreement (Fig. 4d,i). Maximum surface chlorophyll in the observations (> 10 mg m$^{-3}$) and in the simulation (~ 8 mg m$^{-3}$) also match reasonably well. However, the cross-shore and vertical gradients of surface chlorophyll reveal notable differences between the observations and the simulation: local maxima of up to 10 mg m$^{-3}$ are present along the nutricline and at the upwelling front located more than 100 km offshore in the observations, while chlorophyll concentrations in the simulation show no such maxima, are inversely related with SST and decrease almost continuously offshore and with depth (Fig. 3; 4d,i). Lastly, it is a common feature in satellite images of chlorophyll that concentrations remain relatively low (< 1 mg m$^{-3}$) in recently upwelled waters near the coast (30 km) and only increase to > 3 mg m$^{-3}$ further offshore (Fig. 3c). This behaviour is to some degree reproduced in the simulation, but only within a much narrower (~ 10 km) region along the coast (Fig. 3d). The observed nutricline - here defined as the 10 $\mu$mol l$^{-1}$ nitrate contour - is located between 20 m - 50 m depth in the open ocean and intersects the surface near the coast where upwelling occurs (Fig. 4e). The modeled nutricline is locally 100 m deep in the open ocean and also reaches the surface near the coast (Fig. 4j). Surface nitrate maxima of 5 $\mu$mol l$^{-1}$ associated with filaments in the simulation are comparable to the observations.

**In brief, the near-surface cross-shore gradients of temperature, nitrate and chlorophyll are well represented in the simulations. In the following section we will see how both observed and modelled cold filaments give rise to along-shore variability by advection across these gradients.**

**3.3   Physical and biogeochemical characterization of observed and modeled filaments**

In the observations, cold filaments dominate the along-shore variability of physical and biogeochemical parameters near the surface (Figs. 3a,c; 5a-e). Two cold filaments with temperatures of 21.5 °C and 20.5 °C in their respective centers extend offshore from the upwelling center, separated by a ~ 30 km wide intrusion of 1 °C warmer water between them (Fig. 3a). **Their along-shore position matches with two SST minima (19 °C) at the coast, suggesting that they carry recently upwelled**

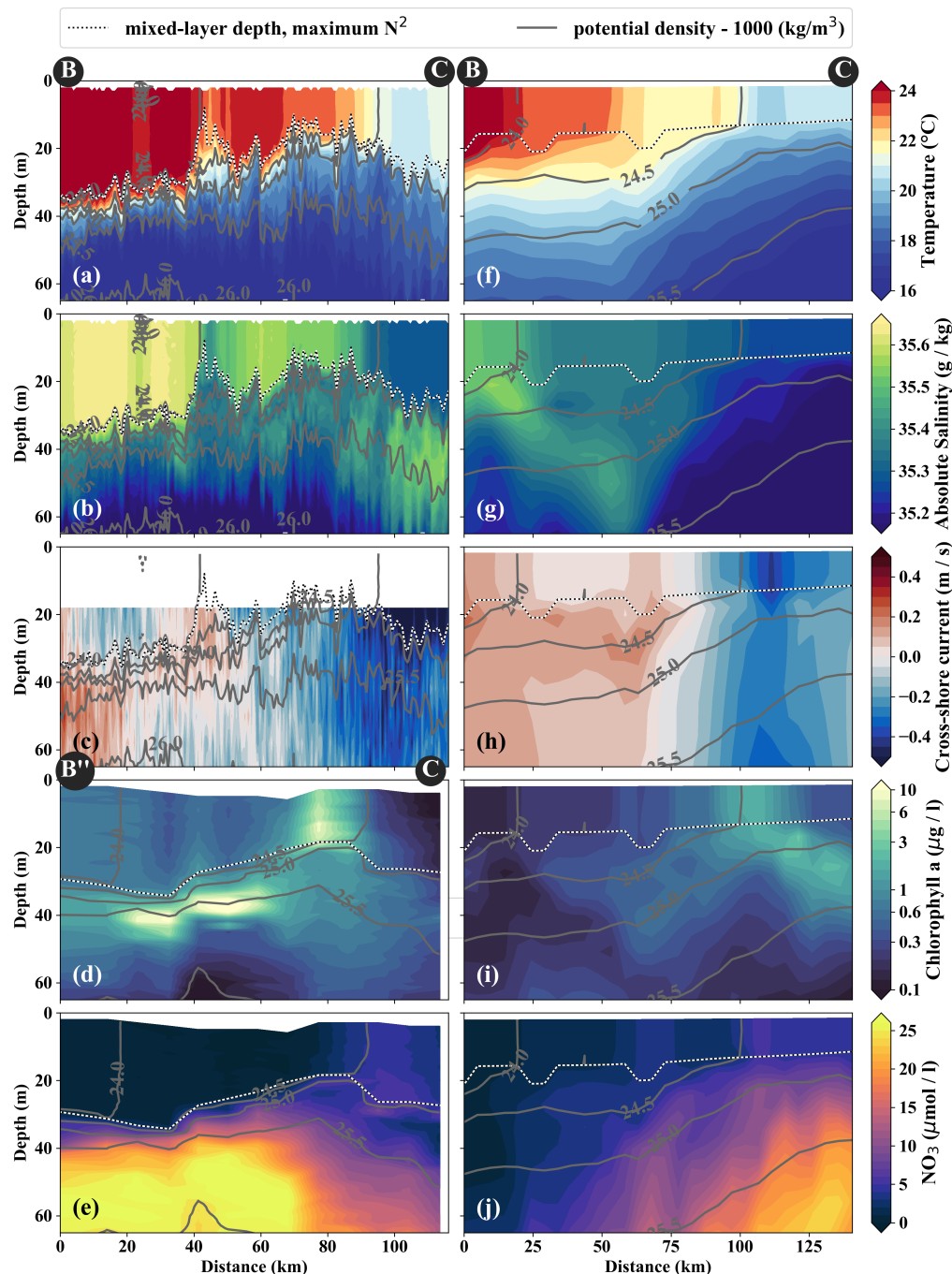

**Figure 5.** Same variables as in Fig. 4. Note the slightly different endpoints of the physical and biogeochemical section in the observations (see B and B" in Fig. 3). **The model sections represent 1-day averaged model output on April 14, 2017, which was chosen for the the horizontal and vertical gradients to be as sharp as possible and directly comparable to observations.**

**water**. In the following we focus on the relatively narrow (10 - 20 km) northern filament at 15.25 °S, 77 °W because of multiple available physical and biogeochemical measurements. The filament can be identified in satellite SST images already on March 22 and changed its position only by $\mathcal{O}(10)$ km until it was sampled on April 15 (not shown). The associated SST fronts are present the entire time, but vary in strength. The physical structure of the filament and the distribution of the biogeochemical parameters **are** characterized in the following.

The cold filament is associated with along-shore variability of the physical and biogeochemical fields in the mixed layer (Fig. 5a-e). It is characterized by a pronounced minimum in temperature (20 °C) and salinity (35.2 g kg$^{-1}$) in the mixed-layer at the southern end (110 km) of transect PHY (Fig. 5a,c). The minimum temperature found in the filament on transect PHY is at least 1.5 °C colder than suggested by the satellite SST (Figs. 3a, 5a). This mismatch is likely related to the diurnal cycle of solar insolation and differential heating, as PHY crossed the filament in the early morning but the SST image was recorded the day before shortly after noon. The low salinity is characteristic of ESSW along the shelf (see Sec. 3.2) and thus indicates that the filament contains recently upwelled water which is transported to the open ocean by an offshore flow of up to 0.5 m s$^{-1}$ within the mixed-layer (Figs. 5c). The subsurface flow is mainly offshore at the southern end of transect PHY as opposed to onshore flow at its northern end, which is related to a cyclonic mesoscale eddy (not shown). Weak stratification below the filament between the 24.5 kg m$^{-3}$ and 25 kg m$^{-3}$ isopycnals (not shown) points to water that has been in the mixed-layer recently and could indicate subduction by submesoscale frontal processes. Low salinity anomalies (35.3 g kg$^{-1}$) in the same density range below the filament support this hypothesis (Fig. 5b).

Along with the physical properties, the filament creates along-shore variability of the biogeochemical parameters (nitrate, chlorophyll) by advecting recently upwelled water into the open ocean (Figs. 5d,e). Nutrient concentrations in the mixed-layer are enhanced in the filament (4 - 7 $\mu$mol l$^{-1}$ NO$_3$) compared to the surrounding waters ($<$ 1 $\mu$mol l$^{-1}$ NO$_3$) while the highest nitrate concentrations are found near the filament's northern edge (Fig. 5e). **A local NO$_3$ maximum is located at the base of the mixed-layer (Fig. 5e).** Despite available nutrients chlorophyll concentrations are very low ($<$ 0.1 mg m$^{-3}$) within the filament, comparable to those found below the photic zone (Fig. 5d). PP in the filament is still relatively high with a maximum (7.5 $\mu$mol C l$^{-1}$ d$^{-1}$) at 10 m depth within a 35 m deep mixed-layer (Table 1). High chlorophyll concentrations ($\sim$ 8 mg m$^{-3}$) are only found at the northern edge of the filament 75 km along transect BIO (Fig. 5d). Outside the filament, surface waters are nutrient-depleted ($<$ 0.2 $\mu$mol l$^{-1}$) while just below the mixed-layer high nutrient concentrations (25 $\mu$mol l$^{-1}$ NO$_3$) are present (Fig. 4e). The maxima in chlorophyll ($>$ 10 mg m$^{-3}$) and PP (9.4 $\mu$mol C l$^{-1}$ d$^{-1}$) are therefore located below the mixed-layer ($\sim$ 40 m) where nutrients are abundant (Fig. 5d,e; Table 1). Below 80 m depth, chlorophyll concentrations are low everywhere along the transect ($<$ 0.1 mg m$^{-3}$) and primary production is low ($<$ 0.1 $\mu$mol C l$^{-1}$ d$^{-1}$) at the upwelling front, offshore and in the filament (Figs. 3c, 4d; Table 1). Notably, surface PP in the filament is a factor 2 higher than at the upwelling front (3.6 $\mu$mol C l$^{-1}$ d$^{-1}$), while the latter dominates the offshore chlorophyll variability in satellite images with surface chlorophyll concentrations of about a factor 4 higher than in the filament (Fig. 3c).

**The position and shape of simulated filaments is determined largely by the mesoscale eddy field, which evolves freely in the simulation and can therefore not be expected to correspond to the variability in the real ocean at any given time. The occurrence of major upwelling events and their effect on the variability of fronts and filaments, however, is closely**

related to the wind forcing of the model, which was derived from satellite-based, daily ASCAT scatterometer winds. We therefore picked simulated filaments that were as close as possible in space and time to the observations, which were then taken to be representative of the area and season when the observations were carried out. The chosen filaments are comparable in scale to the observed cold filament, which had an offshore extent of 150 km - 200 km (Fig. 3a,b). Similar to satellite images, the simulated surface fields of SST and chlorophyll show two separate filament structures originating in the upwelling patches off Pisco and Nazca on April 5, 2017 (Fig. 3b,d). Of the two separate filament structures that are present, for comparability we focus here on the northern filament whose location near the upwelling front is similar to the one observed. The modelled filament is associated with pronounced along-shore variability in the physical and biogeochemical fields (Fig. 5, f-j). Offshore velocities are present in the filament down to 100 m depth, with a maximum located in the mixed-layer (0.4 m s$^{-1}$, Fig. 5h). Surface minima of temperature (20 °C) and salinity (35.25 g kg$^{-1}$) and maxima of chlorophyll and nitrate are associated with the filament (Fig. 5f-j). Enhanced nitrate concentrations of 5 $\mu$mol l$^{-1}$ are present in the filament and a shallowing of the nutricline is found in the same location, associated with doming isopycnals between $\sigma_\theta = 25$ kg m$^{-3}$ and $\sigma_\theta = 25.5$ kg m$^{-3}$ (Fig. 5j). Elevated chlorophyll concentrations of 2 mg m$^{-3}$ are found at its northern edge, marked by the $\sigma_\theta = 24.5$ kg m$^{-3}$ isopycnal outcrop (Fig. 5i). **PP in the modelled filament is enhanced relative to the surrounding offshore waters, similarly to observations (Table 1).** However, the modelled PP is only enhanced where high chlorophyll concentrations are found, which is not the case for the observations where high PP inside the filament coincides with low chlorophyll. Vertical gradients of PP are also different in the model compared with observations: While there is a strong chlorophyll maximum below the mixed-layer that is associated with higher PP than at the surface, subsurface maxima of chlorophyll are rare and weak in the simulation and PP generally decreases with depth. Despite these differences, the modelled mixed-layer PP has a realistic order of magnitude offshore, at the upwelling front and in the filament (Table 1).

**In summary, the simulation exhibits upwelling filaments that are similar in lateral and horizontal scale and offshore extent to those observed. Our direct rate measurements indicate that PP in the modelled filaments is comparable to observations. Despite very low chlorophyll concentrations (< 0.1 mg m$^{-3}$) in the observed filament, surface PP is by a factor 2 higher than at the upwelling front. This highlights the necessity of measuring PP in addition to chlorophyll for model validation to ensure a realistic representation of the underlying processes.**

### 3.4 Timescales of nutrient transport and uptake in observations and models

**The model is able to reproduce realistic mixed-layer nitrate concentrations as well as mixed-layer PP rates offshore in the filaments, suggesting that the simulated rate of nitrate uptake (i.e. new production) is realistic as well (Figs. 4e,j; 5e,j; Table 1). However, it must be noted that our PP in situ estimates do not discriminate between new PP and regenerated PP resulting from ammonium uptake (see discussion section). Assuming that the simulated nitrate uptake rate is realistic, we can use the model to analyze the physical and biogeochemical processes in the upwelling system in more detail and over a larger area.** To do this, we released virtual floats in our submesoscale-permitting simulation. By tracking biogeochemical variables along float trajectories, we studied the biological response to upwelling and subduction in a

**Table 1.** Observed and modelled primary production for different dynamical regimes, each at 3 different depths. The first sample was always taken at 10 m, while the remaining two were placed in the chlorophyll maximum below the mixed-layer (25 m - 40 m, ~ 10% surface PAR) and a chlorophyll minimum below the photic zone (70 m - 80 m, ~ 1% surface PAR). Sampling sites and the corresponding model locations used for comparison are marked with letters O, U and F in Fig. 3c,d. Standard deviation represents triplicate samples for the PP observations, ±3 m bottle depth for the corresponding chlorophyll fluorescence profiles and differences between individual grid points for the model data.

| Regime | Observations | | | Model | | |
|---|---|---|---|---|---|---|
| | Depth (m) | Chlorophyll ($mg\,m^{-3}$) | Primary production ($\mu mol\,C\,l^{-1}\,d^{-1}$) | Depth (m) | Chlorophyll ($mg\,m^{-3}$) | Primary production ($\mu mol\,C\,l^{-1}\,d^{-1}$) |
| Offshore | 10 | $0.45 \pm 0.10$ | $2.12 \pm 0.044$ | 10 | $0.24 \pm 0.038$ | $1.01 \pm 0.134$ |
| | 35 | $7.64 \pm 3.922$ | $9.43 \pm 1.402$ | (10% PAR) 40 | $0.15 \pm 0.009$ | $0.19 \pm 0.033$ |
| | 70 | $0.11 \pm 0.012$ | $0.09 \pm 0.031$ | (1% PAR) 82 | $0.39 \pm 0.029$ | $0.08 \pm 0.005$ |
| Upwelling Front | 10 | $3.49 \pm 0.302$ | $3.59 \pm 0.721$ | 10 | $2.80 \pm 1.348$ | $5.74 \pm 1.925$ |
| | 40 | $7.42 \pm 1.299$ | $5.84 \pm 0.757$ | (10% PAR) 18 | $2.52 \pm 1.528$ | $1.99 \pm 1.016$ |
| | 80 | $0.13 \pm 0.025$ | $0.08 \pm 0.005$ | (1% PAR) 35 | $1.90 \pm 1.307$ | $0.30 \pm 0.192$ |
| Filament | 10 | $0.66 \pm 0.051$ | $7.50 \pm 1.418$ | 10 | $1.43 \pm 0.322$ | $3.78 \pm 0.672$ |
| | 25 | $0.10 \pm 0.016$ | $0.25 \pm 0.021$ | (10% PAR) 22 | $1.07 \pm 0.353$ | $1.14 \pm 0.434$ |
| | 80 | $0.00 \pm 0.002$ | $0.03 \pm 0.001$ | (1% PAR) 47 | $0.22 \pm 0.014$ | $0.04 \pm 0.002$ |

reference frame following the upwelled water. The upwelled floats originate at depths between 20 - 120 m, the maximum depth agreeing well with the maximum depth of outcropping isopycnals (Figs. 4; 6a). Using the physical and biogeochemical properties averaged over all floats after grouping into subducted and not subducted ones, we can diagnose the biological response to upwelling and the corresponding timescale (Fig. 6a-e). **Phytoplankton growth in PISCES increases exponentially with**

5 **photosynthetically active radiation (PAR) and also increases as a function of temperature (subsection 2.5, equation 5).** PP and chlorophyll along the float trajectories thus increase exponentially a few days before upwelling as the floats move to shallower depths where both PAR and temperature are higher and phytoplankton biomass accumulates (Fig. 6a-c). For the floats that are not subducted and remain within the photic zone chlorophyll peaks 15.3 days after upwelling ($4.1\,mg\,m^{-3}$), followed by a peak in PP after 17.1 days ($13.1\,\mu mol\,C\,l^{-1}\,d^{-1}$). For the subducted floats these timescales are slightly shorter,

10 chlorophyll peaks after 14.5 days ($2.0\,mg\,m^{-3}$) and PP peaks after 13.4 days ($4.0\,\mu mol\,C\,l^{-1}\,d^{-1}$).

The fate of upwelled floats is closely related to their change in temperature after upwelling, thus the correct representation of the temperature gradient in the model is crucial (Fig. 3a,b). Water that is warmed rapidly by surface heat fluxes or mixed with warmer offshore waters remains at shallower depths where sufficient light allows for PP and nutrient uptake. In contrast, floats with little temperature change after upwelling are more quickly removed from the photic zone by subduction.

Average PP of the subducted floats reduces to 0.1 $\mu$mol C $l^{-1}$ $d^{-1}$ after 20 days, because they are all located below the photic zone at this time by design of our diagnostics (Fig. 6b). In contrast, average PP of the floats that are not subducted is still at 6.4 $\mu$mol C $l^{-1}$ $d^{-1}$ after 20 days. Nitrate concentrations of subducted floats reduce to 16.5 $\mu$mol $l^{-1}$ which is 1.5 $\mu$mol $l^{-1}$ lower than upwelled concentrations (Fig. 6e). These numbers show that along the subducted trajectories only a small fraction

of the upwelled nitrate is taken up by phytoplankton. In contrast, along the trajectories of those floats that remain in the photic zone phytoplankton can utilise more nitrate indicated by nitrate concentrations of 12.0 $\mu$mol $l^{-1}$ 20 days after upwelling, which is 7.1 $\mu$mol $l^{-1}$ lower than upwelled concentrations. **Using the subduction ratio (Equation 7) defined in subsection 2.5, we estimate that 20 days after being upwelled 40.6% of the upwelled nitrate is subducted (Fig. 6f). More detailed statistics of the float experiment are given in Table 2.**

Nitrate concentrations in the observed filament around 150 km offshore are only 20 - 50% of those measured near the coast and decrease continuously with offshore distance on cross-shore transect CROSS (Fig. 3e). This suggests that upwelled nitrate fuels a substantial amount of the observed PP. To estimate a timescale over which the observed nitrate uptake occurs, we use offshore velocities of 0.3 - 0.5 m $s^{-1}$ observed in the filament which leads to an advection time of 3.5 - 5.8 days from the upwelling center to reach the filament 150 km offshore (this timescale represents a lower bound since the actual path that

the water parcel took is unknown and likely longer than a straight line). Initial nitrate concentrations of 15 $\mu$mol $l^{-1}$ in the upwelled water and 5 $\mu$mol $l^{-1}$ in the filament yield a nitrate reduction of 10 $\mu$mol $l^{-1}$ over this time period, which is well within the uncertainty of the modelled estimate.

### 3.5   Effect of resolution on the biogeochemistry of the Peruvian upwelling

With the purpose of quantifying the impact of the different dynamics at submesoscale-permitting (1/45 $^{\circ}$) resolution on upwelling and subduction of nutrients, we repeated the above float experiment using the mesoscale (1/9 $^{\circ}$) model flow field and compared both results (Table 2). In the mesoscale experiment, the mean nitrate concentration of upwelled floats is **2.4 $\mu$mol $l^{-1}$** higher than in the submesoscale experiment. For the subducted floats, this higher initial nitrate concentration is overcompensated by a higher uptake along the float trajectories so that offshore nitrate concentrations are 2 $\mu$mol $l^{-1}$ lower in the mesoscale

simulation. For the floats that were not subducted, increased nitrate uptake in the mesoscale simulation also overcompensates the initially higher concentrations, so that offshore nitrate concentrations are 5 $\mu$mol $l^{-1}$ lower in the 1/9 $^{\circ}$ compared to the 1/45 $^{\circ}$ simulation. This higher nitrate uptake for both subducted and not subducted floats in the 1/45 $^{\circ}$ simulation can be explained by on average deeper float trajectories (Fig. 7) and therefore lower light levels and lower PP compared to the 1/9 $^{\circ}$ simulation. Accordingly, subduction of upwelled nitrate is increased by 13.9 % in the 1/45 $^{\circ}$ simulation compared to the 1/9 $^{\circ}$

simulation, while the maxima of PP and chlorophyll along the averaged trajectories are reduced by 5.3 $\mu$mol C $l^{-1}$ $d^{-1}$ and 2.2 mg $m^{-3}$, respectively (Table 2).

This increased subduction in the 1/45 $^{\circ}$ simulation is also apparent in the vertical float distribution 20 days after upwelling (Fig. 7). The vertical maximum of the float distribution is located in the top 5 m for the both simulations, but below this surface peak the distributions are very different: In the 1/45 $^{\circ}$ simulation the majority of floats is evenly distributed between 5 - 50 m

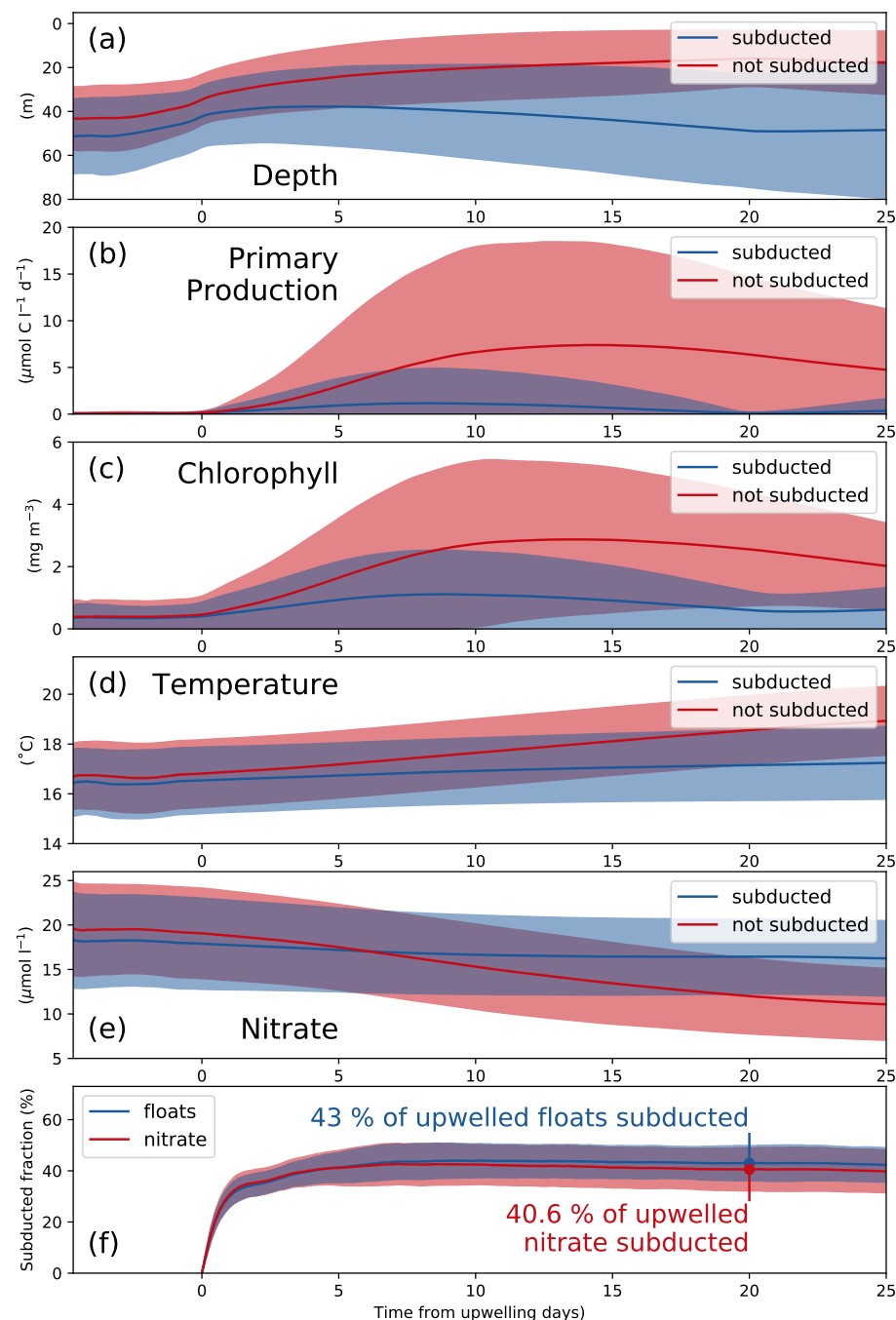

**Figure 6.** (a) Depth, (b) primary production, (c) chlorophyll, (d) temperature and (e) nitrate averaged over all subducted (solid) and not subducted (dashed) floats from 20 ensemble members. Blue and red shading indicates the range of $\pm 1$ standard deviation based on all subducted and not subducted floats, respectively. (f) Subducted fraction of floats and nitrate. Shaded areas represent the range of $\pm 1$ standard deviation based on 20 ensemble members.

**Table 2.** Diagnosed variables from the virtual float ensembles in the 1/45$^\circ$ and 1/9$^\circ$ simulations. "Peak" values represent the maximum of the average over floats of one simulation.

| Parameter | | 1/45$^\circ$ | 1/9$^\circ$ | mean difference |
|---|---|---|---|---|
| Upwelled floats (absolute) | | 5048.9 $\pm$ 1108.0 | 3619.7 $\pm$ 964.1 | -1429.3 |
| Subducted floats (absolute) | | 2158.0 $\pm$ 559.8 | 1149.9 $\pm$ 349.3 | -1008.1 |
| (relative) | | 43.0 $\pm$ 6.9 % | 32.0 $\pm$ 5.7 % | -10.9 % |
| Upwelled NO$_3$ concentration ($\mu$mol l$^{-1}$) | | 17.5 $\pm$ 1.4 | 19.8 $\pm$ 0.8 | +2.4 |
| Subducted NO$_3$ (relative) | | 40.6 $\pm$ 8.4 % | 26.7 $\pm$ 6.0 % | -13.9 % |
| Floats out of domain (absolute) | | 48.9 $\pm$ 34.0 | 0 | -48.9 |
| (relative) | | 1.0 $\pm$ 0.8 % | 0 | -1.0 % |
| Peak primary production | - subducted floats | 4.0 $\pm$ 1.2 | 3.2 $\pm$ 1.0 | -0.8 |
| ($\mu$mol C l$^{-1}$ d$^{-1}$) | - not subducted floats | 13.1 $\pm$ 2.5 | 19.7 $\pm$ 3.3 | +6.6 |
| | - all floats | 9.1 $\pm$ 1.7 | 14.5 $\pm$ 3.0 | +5.3 |
| Peak chlorophyll | - subducted floats | 2.0 $\pm$ 0.4 | 2.6 $\pm$ 0.7 | +0.6 |
| (mg m$^{-3}$) | - not subducted floats | 4.1 $\pm$ 0.6 | 6.6 $\pm$ 0.8 | +2.6 |
| | - all floats | 3.1 $\pm$ 0.4 | 5.3 $\pm$ 0.7 | +2.2 |
| Time of peak primary production | - subducted floats | 13.4 $\pm$ 1.4 | 13.3 $\pm$ 2.5 | -0.1 |
| (days after upwelling) | - not subducted floats | 17.1 $\pm$ 0.9 | 16.1 $\pm$ 1.9 | -1.0 |
| | - all floats | 15.4 $\pm$ 1.0 | 15.2 $\pm$ 1.8 | -0.2 |
| Time of peak chlorophyll | - subducted floats | 14.5 $\pm$ 1.3 | 14.3 $\pm$ 3.1 | -0.2 |
| (days after upwelling) | - not subducted floats | 15.3 $\pm$ 1.4 | 14.3 $\pm$ 2.0 | -1.0 |
| | - all floats | 14.9 $\pm$ 1.1 | 14.3 $\pm$ 2.2 | -0.6 |

depth, while in the 1/9$^\circ$ simulation the float abundance sharply decreases downward from a much more pronounced surface peak. The maximum depth reached by float trajectories is deeper by about 30 m in the 1/45$^\circ$ simulation (Fig. 7). **Comparing the same virtual float experiment in different simulations illustrates well the relative changes in subduction and their biogeochemical impacts, but this Lagrangian approach is restricted to short integrations during one season and is less**

5 **suited to obtain estimates of absolute changes of the time-averaged model fields.**

The impact of increased horizontal resolution on the long-term averaged biogeochemical fields can be approximated based on the difference between the coarse- (1/9$^\circ$) and high-resolution (1/45$^\circ$) simulations. In a first-order assessment, this difference can be taken to represent the influence of submesoscale frontal processes and the associated horizontal and vertical

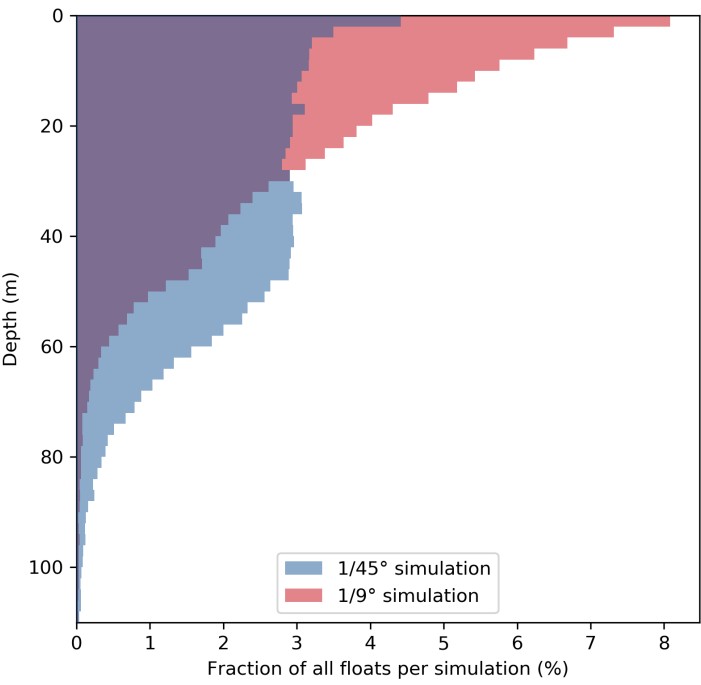

**Figure 7.** Vertical distribution of all upwelled floats from the float experiment ensemble 20 days after upwelling in the 1/9° and 1/45° simulations. Histograms were generated using 2 m vertical bins.

eddy-fluxes that are additionally resolved in the high-resolution simulations, an assumption that is justified by the pronounced increase in eddy kinetic energy by a factor of 2 from coarse to high resolution (Fig. 8a). The changes in nitrate due the effect of submesoscale-permitting resolution in the simulations are greatest within 25 km from the coast in the upper 200 m of the water column, where a pronounced decrease in nitrate ($-2.5\ \mu\mathrm{mol\,l}^{-1}$, Fig. 8b) is apparent. These reduced subsurface nitrate concentrations in the 1/45° compared to the 1/9° simulation are consistent with ~ $2\ \mu\mathrm{mol\,l}^{-1}$ lower nitrate concentrations of upwelled floats (Table 2). **In a tongue that extends from this minimum ~ 250 km offshore along the nutricline, a smaller decrease in nitrate (~ $-1.5\ \mu\mathrm{mol\,l}^{-1}$, Fig. 8b) is seen. Offshore of this negative change, an increase in nitrate (~ $1.5\ \mu\mathrm{mol\,l}^{-1}$) is apparent around 100 m depth. The positive change in nitrate offshore together with increased eddy kinetic energy (EKE) in the 1/45° simulation suggests that an increase in the cross-shore nitrate flux takes over once the nitrate is removed from the surface (Fig. 8a,b). The horizontal patterns of nitrate change are largest in regions of increased eddy kinetic energy (not shown). At a depth of 300 m to 500 m nitrate increases near the coast (~ $2\ \mu\mathrm{mol\,l}^{-1}$), directly below the strong nitrate decrease in the top 200 m (~ $-2.75\ \mu\mathrm{mol\,l}^{-1}$). A plausible explanation for this pattern is that due to increased offshore transport in the submesoscale simulation sinking organic matter sediments on deeper shelf areas,**

**out of reach of the upwelling source waters. This would result in a downward redistribution of nitrate similar to the "nutrient leakage" described in Gruber et al. (2011). However, small differences in the alongshore mean circulation along the continental slope (not shown) may also modify the nitrate content of the water masses. Disentangling these processes is beyond the scope of the present work.** A decrease in chlorophyll of $1 \, \mathrm{mg \, m^{-3}}$ near the coast and $0.3 \, \mathrm{mg \, m^{-3}}$ 300 km offshore between the $1/45\,^\circ$ and $1/9\,^\circ$ simulations is seen at the surface (Fig. 8d). This decrease in surface chlorophyll coincides with an increase in surface nitrate ($+0.5 \, \mu\mathrm{mol \, l^{-1}}$), pointing to reduced nitrate uptake due to limitation of phytoplankton growth by nutrients other than nitrogen. In order to understand the reason for this model behaviour, we diagnosed which nutrient is limiting primary production in the simulation for the surface layer at every grid point and time step. We have identified a decrease of surface iron concentrations by 50% in the $1/45\,^\circ$ simulation (Fig. 8c), leading to a shift from nitrogen to predominantly iron limitation in a 50 km wide band close to the coast over most of the year (not shown). Given that the coastal area is where most production occurs, increased subduction of upwelled iron in addition to nitrate subduction in the $1/45\,^\circ$ simulation could contribute to the reduced productivity at the surface.

## 4 Discussion

Using an integrated approach of in situ observations and modelling, we estimated that 40.6 % of the upwelled nitrate is subducted without being utilised by phytoplankton. **While the agreement between observed and modelled synoptic variability introduced by filaments and fronts lends some confidence to our results, there are uncertainties and limitations which will be discussed in the following.**

We use the fraction of all upwelled floats that **have been subducted 20 days after upwelling** for quantifying changes in the associated nutrient fluxes. These float trajectories represent only advective fluxes while diffusion and surface-mixing are not resolved, yet at submesoscale-permitting resolution and for short integration times it is reasonable to assume that the advective fluxes are dominant and our approach is justified. **A mean bias of the physical and biogeochemical fields in the $1/45°$ simulation (see section 3.1) could have an effect on the results of our Lagrangian analysis. The $17°$C isotherm is steeper near the coast in the $1/45°$ simulation compared to the CARS climatology (Fig. 2c,e), resulting in a negative SST bias of ~ $-3\%$ (Fig. 2a, Fig. 3a,b) that is common in ROMS simulations (e.g. Echevin et al., 2008). More importantly for our study, we identified a positive nitrate bias of ~ 61 % at the surface and ~ 17 % at 100 m depth relative to the CARS climatology (Fig. 2a). A positive nitrate bias of similar magnitude ($6 \, \text{-} \, 8 \, \mu\mathrm{mol \, l^{-1}}$) has also been documented by Espinoza-Morriberon et al. (2017) in their regional PISCES simulations off Peru. The much higher surface bias compared to 100 m depth can be explained by the steeper isotherms near the coast where the nitrate bias is largest, since the nitracline approximately coincides with the thermocline (Fig. 2c,e). The positive nitrate bias suggest an overestimation of the upwelling nitrate fluxes. Since the nitrate concentrations in the coastal upwelling are generally high and thus not limiting PP, it is likely that the nitrate bias does not have a substantial effect on the uptake of nitrate due to PP before subduction. PP in the simulation was also compared to synoptic in-situ measurements (Table 1) and mean satellite estimates (Fig. 2a), and the simulated filaments still contain ~ $5 \, \mu\mathrm{mol \, l^{-1}}$ nitrate offshore which is**

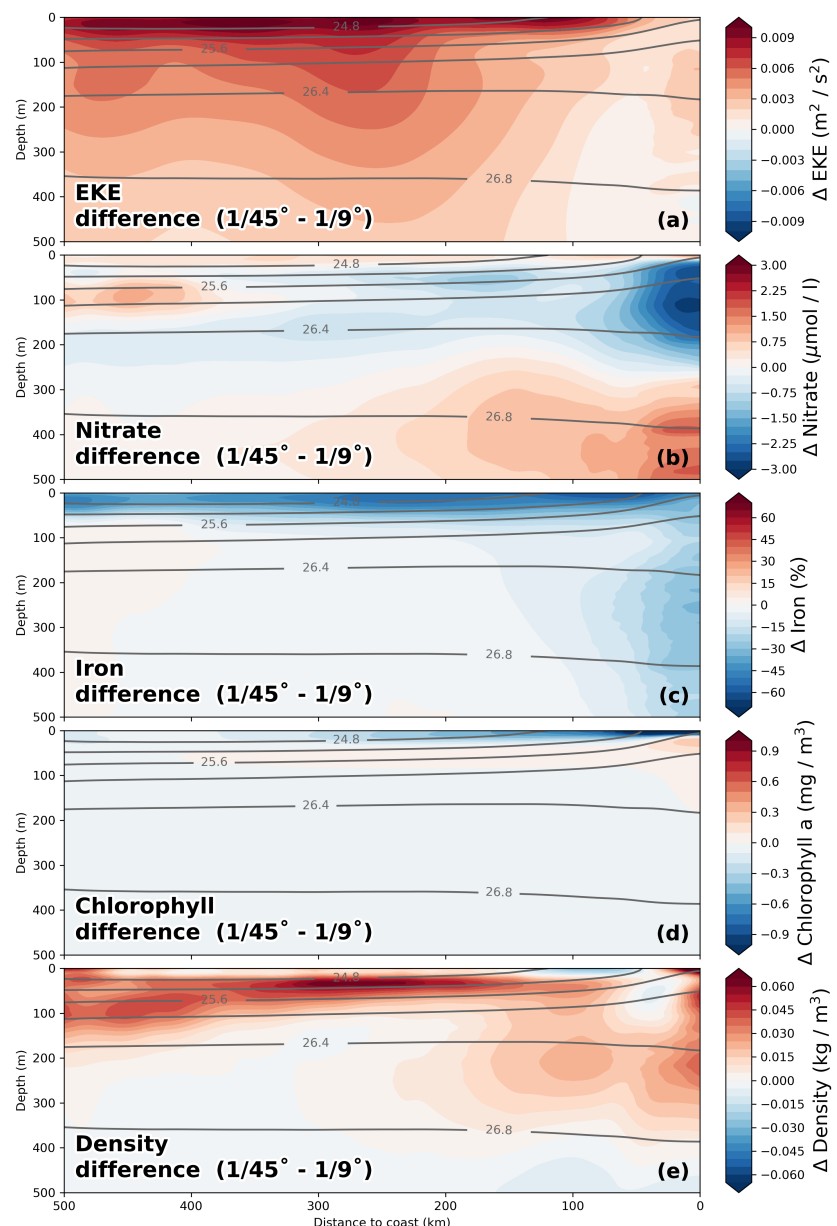

**Figure 8.** Relative changes in (a) Nitrate (b) Iron (c) Chlorophyll (d) Temperature and (e) eddy kinetic energy (EKE) between the $1/9°$ and $1/45°$ simulations calculated as $\Delta X = X_{1/45} - X_{1/9}$. EKE was calculated relative to a 6 month running mean. All fields were averaged on depth levels in along-shore direction with 25 km cross-shore bins in a coastal band of 500 km width between $10.5°S$ and $17.5°S$ over the period 2015 - 2016. Grey lines represent potential density averaged over the $1/9°$ and $1/45°$ simulations.

confirmed in our observations. This further increases our confidence that the nitrate uptake in the simulation is realistic but occurs within a positively biased absolute value range.

The nitrate bias, however, can still impact our estimate of the subducted nitrate fraction (Equation 7, Fig. 6f). To obtain a first-order estimate of the uncertainty due to this bias, we compute the subduction ratio again from averaged mean quantities along the float trajectories. In this case equation (7) can be simplified to

$$\text{ratio} = \frac{F_{\text{subducted}}}{F_{\text{upwelled}}} \left( 1 - \frac{\langle \Delta \text{NO}_3 \rangle_{\text{subducted}}}{\langle \text{NO}_{3,t_0} + B_{\text{NO}_3} \rangle_{\text{subducted}}} \right) \tag{8}$$

where $F$ is the respective number of upwelled and subducted floats and angled brackets denote averages over all subducted float trajectories. The subducted float fraction $F_{\text{subducted}}/F_{\text{upwelled}}$ is 43% in our study and not affected by the nitrate bias. The second term in the parentheses reduces this subducted float fraction, which explains why the subducted nitrate fraction (red line in figure 6f) is always lower than the subducted float fraction (blue line in figure 6f). The reducing term represents the ratio between the nitrate uptake after upwelling $\Delta \text{NO}_3$ and the sum of upwelled nitrate concentration $\text{NO}_{3,t_0}$ and nitrate bias $B_{\text{NO}_3}$. Because the nitrate bias occurs in the denominator, it leads to an overestimation of the subducted nitrate fraction. To approximate this error, we insert averaged values from our float experiments (Fig. 6e) into equation (8): For the ensemble mean values of our float experiment ($\Delta \text{NO}_3$=1.5$\mu$mol l$^{-1}$, $\text{NO}_{3,t_0}$=16.5$\mu$mol l$^{-1}$) and no bias we obtain a subducted nitrate fraction of 39%, close to our original estimate (40.6%, Fig. 6f). Correcting for an assumed positive nitrate bias of 65% ($\Delta \text{NO}_3$=1.5$\mu$mol l$^{-1}$, $\text{NO}_{3,t_0}$=10$\mu$mol l$^{-1}$) we obtain a subducted float fraction of 36.5%. This calculation still assumes that $\Delta \text{NO}_3$ is unbiased, but in reality there may be a negative bias due to the ~ $-37\%$ negative bias of PP described above. Assuming that $\Delta \text{NO}_3$ also exhibits a $-37\%$ negative bias and correcting for it ($\Delta \text{NO}_3$=2.4$\mu$mol l$^{-1}$, $\text{NO}_{3,t_0}$=10$\mu$mol l$^{-1}$) yields a subducted nitrate fraction of 30.5%, which can be viewed as a lower limit of the uncertainty range.

In summary, we approximate that the mean biases of nitrate and PP in the 1/45° simulation likely result in an overestimation of the subducted nitrate fraction with a lower bound of 30.5% of upwelled nitrate being subducted. It should also be noted that not all upwelled floats reach the surface at the time of upwelling but instead originate from depths within the photic zone were the nitrate bias is smaller, which may alleviate the error associated with this bias to some extent. Beyond this, it is possible that nitrate in the CARS climatology is biased low due to undersampling of high-nutrient conditions. Espinoza-Morriberon et al. (2017) suggest that their extensive in-situ nitrate and surface chlorophyll observations off Peru likely still suffer from insufficient sampling resolution.

The physical and biogeochemical observations of the upwelling center near 15°S off Peru agree well with previous studies. Nitrate concentrations of ~ 20 $\mu$mol l$^{-1}$ at the coast and 5 $\mu$mol l$^{-1}$ offshore that were observed in this study are in good agreement with previous measurements in the upwelling near 15°S off Peru and further north on the Peruvian shelf that range between 0.89 - 17.1 $\mu$mol l$^{-1}$ (Blasco et al., 1984; Dengler, 1985; Macisaac et al., 1985; Fernández et al., 2009; Kalvelage

et al., 2013). The observed cold filament described in this study exhibited high nitrate and low chlorophyll concentrations in its core but enhanced chlorophyll at the front separating it from the warm offshore waters (Fig. 5). The same pattern was observed by Hosegood et al. (2017) in a similar filament off Mauritania (150 km offshore extent, 0.4 m s$^{-1}$ offshore velocity), suggesting that these may be general features of EBUS cold filaments.

**The range of PP rates that were determined experimentally from incubations (Table 1) is very close to measurements by Dengler (1985) acquired during April-June 1976 in the upwelling center near 15°S (4 - 16 $\mu$mol C l$^{-1}$ d$^{-1}$), increasing confidence in our results. In October 2005, Fernández et al. (2009) observed a similar range of PP rates in the top 25 m of the water column slightly north (3 - 16.5 $\mu$mol C l$^{-1}$ d$^{-1}$; 11 - 12°S) and south (1 - 3.5 $\mu$mol C l$^{-1}$ d$^{-1}$; 16 - 18°S) of our study area. The PP rates diagnosed in the biogeochemical model simulations are close to our observed PP rates and**
**represent well the horizontal variability between different dynamical regimes in the study region (Table 1), except for the offshore waters where the modelled subsurface PP is likely too low. Satellite-based estimates of mean NPP suggest that the vertically integrated PP in the model has a negative bias (~ −37%, Fig. 2a).**

    **Besides, our observations of PP rates in the filament do not distinguish between regenerated production from ammonium and new production from nitrate (Dugdale and Goering, 1967). Fernández et al. (2009) show that regenerated**
**nitrogen can locally account for as much as 50% of surface primary production in southern Peru. However, nitrate concentrations measured in the filament 150 km offshore are only 20 - 50% of those measured near the coast and nitrate appears to decrease with offshore distance on transect CROSS (Fig. 4). This is an indicator that a substantial amount of PP in the region of the filament survey near 15°S is fuelled by newly upwelled nitrate.** We estimated using observed velocities that a reduction in nitrate concentrations by 10 $\mu$mol l$^{-1}$ occurs over the course of 3.5 - 5.8 days while the water
is advected offshore. Assuming this reduction in nitrate concentrations occurs only due to phytoplankton growth in a closed volume would require PP rates of 11.4 - 18.9 $\mu$mol C l$^{-1}$ d$^{-1}$ (using a C : N ratio of 6.625 after Redfield, 1963). This estimate is higher than the ~ 7.5 $\mu$mol C l$^{-1}$ d$^{-1}$ measured in the filament, possibly pointing to nitrate being removed from the surface waters due to subduction and diffusion processes. Admittedly, large uncertainties are associated with this crude estimate since it is likely that (1) PP in the upwelled water did not remain constant, (2) the ratio of C and N uptake does not correspond
exactly to the Redfield ratio, (3) the water parcel did not travel offshore in a straight line, (4) was subject to mixing with the surrounding water and (5) local remineralisation of nutrients took place. Nevertheless, it provides a useful illustration that advected nitrate from the upwelling patch near the coast is more than sufficient to support the observed phytoplankton growth inside the filament. It is therefore reasonable to assume that a substantial fraction of PP occurring within the filament is fuelled by newly upwelled nitrate.

**Along-shore variability of PP in the simulation is closely related to chlorophyll concentrations, while the observed relationship of PP and chlorophyll is less clear (Table 1). Investigating the reasons for this discrepancy is beyond the scope of this study and left for future work, but we can speculate that it is related to the relatively simple parameterisation of primary production in PISCES (see section 2.5). For example, changes of the phytoplankton physiology or species composition of the phytoplankton community are not modelled, which would likely allow stronger variations**
**of the chlorophyll to carbon ratio of phytoplankton (section 2.5, equation 5). Another missing process is the aging or**

**health of the phytoplankton population, which could slow down PP while chlorophyll concentrations remain high and thereby decouple the two parameters. Note that a new version, PISCES-quota (Kwiatkowski et al., 2018) will be able to represent more elaborate phytoplankton stages in future studies.** Vertically, the modelled PP strictly decreases with larger depth corresponding to lower light levels, while in the observations subsurface maxima of PP are found offshore and at the upwelling front (Table 1). The subsurface chlorophyll maximum is too weak and diffuse in the simulations, which is consistent with this difference in the vertical distribution of PP. This could be partly related to dynamical bias in the simulations: Near-surface stratification at the base of the mixed-layer is by a factor of 2-3 lower in the model than in the observations, which likely results in a too diffuse offshore nutricline and contributes to the weak subsurface chlorophyll maximum. For the reasons mentioned above, subsurface PP in the offshore waters is likely too low in our simulations.

We estimated the impact of submesoscale frontal processes by comparing virtual float experiments in the 1/9° and 1/45° simulations and found that subduction of nitrate increases by 13.9 % at submesoscale-permitting resolution. A decrease in mean PP by about one-third going from 1/9° to 1/45° resolution is also seen along float trajectories (Table 2). The difference between two-year averaged fields shows a decrease of subsurface nitrate concentrations by about 2.5 $\mu$mol l$^{-1}$ within 200 km from the coast in the 1/45° simulation (Fig. 8b), further supporting this interpretation. These results suggest that submesoscale frontal processes amplify the mesoscale effects found in previous studies (Gruber et al., 2011; Nagai et al., 2015) namely reducing PP by enhancing the downward and offshore transport of nutrients and phytoplankton biomass. In an approach similar to ours, Gruber et al. (2011) quantified the net effect of eddy-fluxes by comparing an eddy-permitting (5 km) simulation with a "non-eddy" simulation of similar resolution but with non-linear terms in the model equations set to zero. Comparing the change in nitrate between these simulations (see Fig.3f in Gruber et al., 2011) with the change in nitrate between our 1/9° and 1/45° simulations (Fig. 8b) in our study shows similarities. The patterns of decreased nitrate concentrations within ~ 200 km from the coast and increased nitrate concentrations further offshore (~ 400 - 500 km) are in good agreement (not shown). The patterns of the changes in density also match and can be explained by lateral eddy-fluxes that induce a shoreward heat transport which flattens the isopycnals (Fig. 8d, Gruber et al., 2011). **More generally, Zhong and Bracco (2013) and Zhong et al. (2017) find that submesoscale frontal processes are important for the vertical tracer transports near the surface and that vertical dispersion of Lagrangian particles strongly increases with increasing horizontal model resolution, which is also in agreement with the results of this study (Fig. 7).** Our results further agree with the idealised model results by Lévy (2003) that submesoscale frontal processes export nutrients downward and offshore in regions with relatively high surface nutrient concentrations and can thereby reduce PP. At the same time, this enhanced downward and offshore transport would actively transport fresh organic matter into the oxycline and potentially stimulate microaerobic and anaerobic activity in the upper part of the offshore OMZ (Kalvelage et al., 2013, 2015). **Subduction at fronts has previously been shown in observations to be as important for organic matter export below the photic zone as gravitational sinking (Stukel et al., 2017).**

Note that these previous studies used relatively simple biogeochemical models based on the nitrogen cycle only. Our simulations with a more complex model including the iron cycle suggest that a more complex response of the biogeochemistry to changes in the dynamics is possible. In our simulations, iron limitation occurred offshore over a larger area of the domain and closer to the coast in the submesoscale-permitting simulation (not shown), suggesting that iron may be even more strongly

affected by the subduction associated with submesocale filaments and fronts than $NO_3$. This is in line with recent findings of Browning et al. (2018), who observed iron limitation of phytoplankton growth between 73 km - 266 km from the Peruvian coastline, in contrast with nutrient replete conditions 26 km from the coast on the shelf. The authors conclude that iron is likely an important factor driving reductions in offshore phytoplankton productivity in the PCUS. This suggests that the widespread

occurrence of iron limitation outside the nearshore area (~ 20 km) in our simulations is realistic.

To further investigate the processes driving the iron difference in the surface layer (Fig. 8c), we computed the iron supply from the sediments in the 1/9° and 1/45° simulations. The parameterisation of this sedimentary iron input used in PISCES only depends on water depth as an indication for how well the sediment is oxygenated. Due to slightly different topographies in the 1/9° and 1/45° grids caused by spatial smoothing, the iron supply from near-surface sediments (integrated over the upper 200 m

of the water column that potentially supply the upwelling) is 12.8 % higher in the 1/45° simulation, while the total sedimentary iron input is 22.8 % lower in the 1/45° simulation due to a steeper shelf break. This explains the reduced iron concentrations in the 1/45° simulation at 200 m - 400 m depth near the shelf (Fig. 8c), but does not affect the surface dynamics. This indicates that the strong negative surface iron anomaly does not result from differences in the sediment flux but likely from an enhanced subduction of iron-rich upwelled waters due to submesoscale dynamics.

The Lagrangian analysis using virtual floats allowed us to reduce the complex spatial and temporal variability of coastal upwelling and determine an average biological response to upwelling and subduction. This analysis was effective for gaining insights into physical-biogeochemical interactions from relatively short simulations. Nagai et al. (2015) conducted a virtual float experiment similar to the one in this study. They reported that 30 - 40 % of floats they released in their simulation of the Californian upwelling system were subducted below 50 m after 4 - 6 months. This compares well to our estimate, although

their model is only mesoscale-resolving. This could be explained by differences in the respective regions that are considered: In our study we allow for upwelling inshore of the 25 kg m$^{-3}$ isopycnal which approximately represents the upwelling front. In contrast, Nagai et al. (2015) only released floats in a narrower region within 15 km from the coast where the probability of subduction is higher. Restricting our analysis to a smaller region on the dense side of the 25.5 kg m$^{-3}$ isopycnal, we obtain a 5.9 % larger subducted float fraction for the 1/45° simulation. Other possible reasons for their relatively high subducted

float fraction are that eddies are generally more energetic off California compared to Peru (Capet et al., 2008a) and that they diagnose the subduction of floats over a period of 3 - 4 months compared to only 20 days in our study.

In addition to the aforementioned increase in subduction between the 1/9° and 1/45° simulations, there is also a larger number of floats upwelled in the submesoscale-permitting simulation (Table 2). This is consistent with an increase of mean upwelling velocity on the shelf between 13°S and 16°S in the 1/45° simulation (not shown). Moreover, our results show that

the enhanced upwelling and subduction in the submesoscale simulation do not cancel out but lead to substantial differences in the biogeochemical tracer fields. Using a similar approach of increasing model resolution (1/3° to 1/30°) to evaluate the impact on $CO_2$ air-sea fluxes in the Californian EBUS, Fiechter et al. (2014) also found an upwelling increase with increasing spatial resolution (see their Fig.7) but did not propose an explanation for this particular response. One likely explanation for this difference could be the more accurately represented steep shelf in the southern part of the domain in the 1/45° simulation due

to less topography smoothing. However, the main aim of this study is to deepen the understanding of the subduction process

occuring after the upwelling. Thus it is beyond the scope of this paper to investigate in detail the reasons for the changes in mean upwelling related to refining the coastline and increasing spatial resolution.

## 5 Conclusions

In this study we used an integrated approach combining high-resolution biogeochemial observations and a submesoscale-permitting model simulation to quantify subduction of upwelled nitrate off Peru. Our Lagrangian analysis of the simulations indicates that 40.6 % of the nitrate upwelled near 15°S off Peru inshore of the upwelling front is subducted again and remains unused by phytoplankton, and that 43 % of the upwelled water is subducted. **Taking model biases into account we give a best estimate for subduction of upwelled nitrate off Peru between 30.5 - 40.6 %. Comparing these results with a mesoscale simulation we find that submesoscale frontal processes increase subduction of upwelled water by 10.9 % compared to mesoscale processes, increasing the associated subduction of nitrate by 13.9 % and reducing PP by about one-third.** Average surface chlorophyll concentrations are reduced by up to $1 \, \mathrm{mg \, m^{-3}}$ and subsurface nitrate concentrations are reduced by $2.5 \, \mu\mathrm{mol \, l^{-1}}$ within 200 km from the coast. These results suggest that submesoscale frontal processes amplify the mesoscale effects found in previous studies (Gruber et al., 2011; Nagai et al., 2015) by enhancing the downward and offshore transport of nutrients and fresh organic matter. In addition, surface iron concentrations were lower in the submesoscale-permitting simulation, shifting the coastal ecosystem into iron limitation and reducing PP. This effect could not be seen in previous studies that used simpler biogeochemical models. Mesoscale model studies likely underestimate the reduction of PP due to eddies and filaments.

*Code and data availability.*   All code and data are available upon request. The PYTHON toolbox used for visualisation of the simulations can be downloaded at https://github.com/jaard/xcroco. The lowered CTD (https://doi.pangaea.de/10.1594/PANGAEA.904013), underway CTD (https://doi.pangaea.de/10.1594/PANGAEA.904288) and VMADCP (https://doi.pangaea.de/10.1594/PANGAEA.901425) data collected on R/V Meteor during cruise M136 are available on the PANGAEA platform. Model output cannot be efficiently hosted online due to storage space constraints and energy considerations, but will instead be made available upon request.

*Author contributions.*   J. Hauschildt carried out all analysis, prepared the figures and wrote the main manuscript. S. Thomsen designed and conducted the field experiment on F/S Meteor, during which L. Bristow and G. Lavik performed primary production measurements. V. Echevin and Y.S. Jose both carried out part of the model simulations analysed in the manuscript. V. Echevin, S. Thomsen and A. Oschlies provided guidance and ideas during the analysis and the interpretation of the results. V. Echevin and S. Thomsen participated in writing the manuscript. G. Krahmann performed data processing and calibration including the novel $NO_3$ measurements. All authors commented and reviewed the manuscript.

*Competing interests.* The authors declare that they have no conflict of interest.

*Acknowledgements.* This work is a contribution of the Sonderforschungsbereich 754 Climate-Biogeochemistry Interactions in the Tropical Ocean (www.sfb754.de), which is funded by the Deutsche Forschungsgemeinschaft (DFG). Soeren Thomsen received funding by the European Commission (Horizon 2020, MSCA-IF-2016, WACO 749699: Fine-scale Physics, Biogeochemistry and Climate Change in the West

5  African Coastal Ocean). Yonss S. Jose received funding from the BMBF (Humboldt Tipping Points, 01LC1823B). We are grateful to the Peruvian authorities for the permission to carry out scientific work in their national waters. Special thanks go to the captain and the crew of the R/V Meteor for their support during the M136 cruise and to the chief scientist Dr. Marcus Dengler. For processing the nutrient samples used for the $NO_3$ sensor calibration we would like to thank the group of Dr. Stefan Sommer and their technicians. We would also like to thank Clarissa Karthäuser and Gabriele Klockgether for help with sample analysis for the primary production measurements.

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
