# Peer review of "The fate of upwelled nitrate off Peru shaped by submesoscale filaments and fronts"

_Biogeosciences, 2020_

## Referee Comment (RC1) · Takeyoshi Nagai (Referee) · 1 Jun 2020

This study investigates the effects of filaments and fronts on subduction of nitrate in the Peru-Chile Upwelling System, using a submesoscale permitting regional ocean model coupled with a biogeochemical model. The results of this study found that about 40% of the newly upwelled nitrate near the coast of Peru is subducted before being used by phytoplankton. The results show a more pronounced subduction occurs at submesoscale fronts near the cold filaments, which were underestimated by the previous mesoscale resolving models. The manuscript is well written and includes new insights on the subduction of the nitrate in EBUS. Therefore, I have only several minor comments below.

[Figure]

P1L11 "highlighting the additional value of direct rate measurements for model validation. " Unclear

P2L13 "citepChavez2007, Gruber2015, Koehn2017, Brady2019. " Typo

P11L8 "80m (100m)" Spaces are needed before "m".

P11L29 "30km" A space is needed before "km".

P11L30 "...along-shore position matches with two SST minima (16 âǓę C) at the coast..." I couldn't find two SST minima 16 degree at the coast in Figure 2. From the color scale they look like 18-19 degree.

P14L2 "...is characterized in the following..." "is" should be "are".

Figure 3 and 4 captions There is no explanation for the broken white lines, which are probably MLD by reading legend, but the information should be in the captions.

P17L28 "In brief, in situ PP observations based on carbon suggest that a significant portion of the observed offshore reduction in nitrate concentrations is due to uptake by PP. Simulations indicate that approximately 40.6% of upwelled nitrate is subducted 30 and not utilised by PP. " Is this because PP estimated in-situ without the effect of subduction of nitrate?

P19L32 "While the coincidence of increased subduction and a negative change in nitrate may seem counterintuitive, ..." To me, increased subduction and reduction of nitrate are not counterintuitive.

P20L2 "−2 " Is the minus sign here for a increase?

P20L3 "A plausible explanation for this pattern is that organic matter is able to sediment at shallower depth on the slope in the mesoscale simulation, which permits enhanced local remineralization and nitrate release with respect to the submesoscale simulation. " I'm confused. If this is the case, remineralized nitrate is higher in 1/9 degree simulation with negative value in Figure 7b in below 300 m near the coast?

P23L7 "Along-shore variability of PP in the simulations is closely related to chlorophyll concentrations, while the observed relationship of PP and chlorophyll is clearly more complex (Table 1). Investigating the reasons for this discrepancy is beyond the scope of this study and left for future work " Is this because PP estimated in-situ without the effect of subduction to dark layers?

P23L13 "Near-surface stratification in the model is by a factor of 2-3 lower than in the observations " What mixing parameterization used in this study is missing in the manuscript. Is it KPP or two-equation type turbulence closures?

P23L18-31 "However, the fact that nitrate concentration..." I think that PISCES includes ammonium. Therefore, the authors can see the fraction of ammonium for the PP.

---

## Referee Comment (RC2) · Anonymous Referee #2 · 24 Jul 2020

The authors present a study that combines observations and modeling in order to assess the role of submesoscale filaments and fronts in the lateral and vertical redistribution of the upwelled nutrients. They use ship data collected both along shore and cross shore transects off the coast of Peru to study the structure of upwelling filaments and their role in the nutrient transport and subduction, and its impact on primary production. Further, they use a coupled CROCO+PISCES simulation to strengthen these findings and expand their analysis through lagrangian experiments that allow them to follow the biogeochemical transformations of upwelled particles. They find that submesoscale filaments adevect upwelled nutrients offshore, while also subducting them, therefore limiting production.

The paper is interesting and contains relevant in situ data on these submesoscale

structures. I appreciated the Lagrangian approach to study biogeochemical transformations and the fate of the upwelled particles. The results are interesting: they strengthen previous claims regarding the importance of filaments and fronts for the offshore transport, while clarifying better the role of these structures in the enhancement of subduction (rather than submesoscale upwelling). The writing is good, with a few typos and some potential for improving the paper structure (see comments below).

I have two major comments that I'm listing below, plus a few minor/detailed comments following. My main concerns regard: lack of a thorough model evaluation, lack of a clear desccription of the biogeochemical model and therefore of the meaning of certain processes the authors refer to to draw conclusions. I also have a few concerns regarding the paper structure, which I am listing in the detailed comments as they refer to some specific sections. I suggest that the paper is published after revisions are done.

MAJOR COMMENTS

1) Model evaluation

In my opinion, the paper lacks of a thorough model evaluation. Was this exact model setup and run used in a previous publication, where a full evaluation can be found? Is there an appendix or supplementary material that I missed containing more information in this regard? If this is not the case, I strongly encourage the authors to provide a better evaluation of the model, as the current information doesn't seem enough to me. The only figures that provide some comparison between model and observations are Figure 2, 3, 4. However, these images refer to specific days (April 14 2017 and April 5 2017 respectively in observations and model), and they seem to show pretty large biases in the temperature and salinity fields in the model, especially in the upwelling area. These are accompanied by very high nutrient levels, suggesting a high bias in the upwelling fluxes overall. Both the physical and biochemical tracer biases can have important reprecussions on the lateral and vertical fluxes and transformations which require to be discussed in the paper discussion section as they may impact the results

of the Lagrangian analysis. No longer term mean evaluation of the model is explicitly provided (even though the model was run for several years). Also, the discussion of the model performance in the paper text is too qualitative and scattered throughout the text, while it would be worth having a dedicated subsection. I strongly suggest that the authors include in the paper some difference plots (SST, SSS, MLD, velocities, EKE, CHL, nutrients) and/or use a Taylor diagram to summarize the performance of the model in the region of study in order to be able to assess its limitations in the discussion.

2) Biogeochemical model

Thoughout the text, the authors refer to key biogeochemical processes resolved by the model PISCES. However, these processes are never well defined in terms of what other modeled processes and variables they depend upon, and which quantities drive and regulate them in the model. Sometimes assumptions are made regarding modeled biogeochemical processes, when the way they work should be known from the model equations. Please, include an explicit description of the few relevant model processes in your methods section, and clarify to the reader how they relate to each other, providing fundamental knowledge regarding the characteristics of the employed biogeochemical model. Example 1: in page 15, lines 10-16, the authors discuss the fact that chlorophyll and production relate differently to each other in model and observations; this depends on the model code, which is known, and therefore the question is: how does PP relate to CHLA in the model? Can you explain these differences using the model equations, since they can be checked, and is it worth suggesting any amendement to the processes represented by the model given what you find in the observations? Since this process is of interest, the authors should include more information on the model representation of these processes in the methods. Example 2: page 15, lines 29-30: The authors talk about modeled primary production and nitrate uptake, however no information was provided to understand how the two relate to each other in the model. What processes contribute to nitrate uptake in PISCES?

—————

Introduction/Discussion: Please, include the following relevant literature and discuss how it compares to your findings

FILAMENTS STRUCTURE

- Bettencourt, J. H., Rossi, V., Hernández‐García, E., Marta‐Almeida, M., and López, C. (2017), Characterization of the structure and cross‐shore transport properties of a coastal upwelling filament using three‐dimensional finite‐size L yapunov exponents, J. Geophys. Res. Oceans, 122, 7433– 7448, doi:10.1002/2017JC012700.

IMPACT OF MODEL RESOLUTION ON VERTICAL FLUXES

- Zhong, Y., and Bracco, A. (2013), Submesoscale impacts on horizontal and vertical transport in the Gulf of Mexico, J. Geophys. Res. Oceans, 118, 5651– 5668, doi:10.1002/jgrc.20402.

- Zhong, Y., Bracco, A., Tian, J. et al. Observed and simulated submesoscale vertical pump of an anticyclonic eddy in the South China Sea. Sci Rep 7, 44011 (2017). https://doi.org/10.1038/srep44011

SUBDUCTION BY FRONTS

- Stukel, M.R., Song, H., Goericke, R. and Miller, A.J. (2018), The role of subduction and gravitational sinking in particle export, carbon sequestration, and the remineralization length scale in the California Current Ecosystem. Limnol. Oceanogr., 63: 363-383. doi:10.1002/lno.10636

- Michael R. Stukel, Lihini I. Aluwihare, Katherine A. Barbeau, Alexander M. Chekalyuk, Ralf Goericke, Arthur J. Miller, Mark D. Ohman, Angel Ruacho, Hajoon Song, Brandon M. Stephens, Michael R. Landry Proceedings of the National Academy of Sciences Feb 2017, 114 (6) 1252-1257; DOI: 10.1073/pnas.1609435114
* * *
DETAILED COMMENTS

Page 2, line 10-11: This sentence is too long, please split it in two parts.

Page 2, line 34: Please add the following modeling study for the filament transport in the Canary Upwelling System: Lovecchio, E., Gruber, N., & Münnich, M. (2018). Mesoscale contribution to the long-range offshore transport of organic carbon from the Canary Upwelling System to the open North Atlantic. Biogeosciences, 15(16), 5061-5091.

Page 3, line 6-7: Please rephrase the following sentence "For instance, if nitrate uptake by PP occurred much faster than subduction, then mainly organic matter produced in the surfacelayer would be subducted, whereas if it were slower, nitrate would be subducted. "

Page 3, line 9: What does "process studies" mean?

Page 3, line 23: Is 2.5 km resolution enough to resolve submesoscale? I thought one needed at least 2km, if not 1 km. Could you please refer to literature to confirm your statement what 2.5 km is enough?

Page 3, line 26, point nr.1: As I stated, I don't think the paper answers this question, because the model evaluation in the paper is extremely limited and restricted to a few specific simulated events rather than the mean state of the system and its dynamics. Also this is more of a model evaluation problem than a scientific question. I wouldn't include question nr 1 here, I would only leave questions nr 2 and 3.

Subsection 2.5: Please provide here a description of how the relevant processes discussed in the paper are represented by the biogeochemical model PISCES. Please, specify what is the time resolution of your model output.

Subsection 3.1: The structure of this subsection is confusing, as it alternates results

from the observations (paragraphs 1 and 3) and results from the model which actually in great part is a model evaluation. Please, move out the model evaluation from here a create a proper section that focuses on it, in order to refer to it also in the discussion when considering the limitations of the model. Also, please list first all the results obtained from the observations, and then those obtained from the model, whithout switching from one to the other in each paragraph.

Page 11, line 7: "The variability of biogeochemical fields is similar to observations". Please, avoid using generic terms such as "similar".

Page 11, line 21-24: Please, remove this summary: these results are expected in any upwelling region, so it doesn't really add much information.

Figure 3 and Figure 4: What do the model output sections represent? Is this still a single instance such as Figure 2 (April 5th 2017)? Or is this some longer term mean? Please, state this in the captions. Also, how was the specific day for the model analysis and plots chosen? Currently, some explaination is given in page 14 lines 31-35, however some of the plots have already been introduced before this paragraph. Also, I would assume there are several events that may resemble the observations in the many years of model run, why was only a specific event chosen rather than using an ensemble of suitable events? Please, explain your rationale for the choice of day either in the captions or in the Methods.

Page 14, line 21: The paper provides a discussion of PP and how differently it relates to CHL in model and observations. Therefore, why is PP not plotted in Fig 3 and 4 if it's a variable of interest? Would it make sense to add it?

Page 14, line 31-32: Please, rephrase the first sentence as it's not good English.

Page 14, line 33: Please rephrase here with "We therefore picked simulated filaments..."

Page 15, line 10: Please, substitute similar with similarly

Page 15, line 17-27: This paragraph is not very useful and sounds like a repetition. It could be removed.

Page 15, line 29-30: I don't understand this first sentence. Since one can actually determine the simulated rate of nitrate uptake in the model, why should one "assume" something about it?

Page 17, line 14-24: This section belongs to the Methods, not the Results.

Page 17, line 24-26: Why should this implicit assumption be reasonable? Could you please provide some reason? Also, what is the definition of "subducted" - below what depth (this should be defined in the methods)?

Page 16, line 10: This initial sentence about the importance of SST remarks that it's crucial to include a good evaluation of the temperature gradient in the model.

Figure 7: The x-axis labels and ticks overlap in the subplots.

Page 19, line 1-13: Wouldn't it be necessary to also discuss how the general model performance changes changing the resolution? What about the effects linked to the change in resolution of the forcing such as the wind stress, which is known to influence both upwelling and small scale flow? This paragraph compares two runs, but none of them was thoroughly evaluated.

Page19, line 20: Please add "is" in "during one season and is less..."

Page 19, line 31-32: Please rephrase better the sentence "Further offshore..."

Page 20, line 1: Who are "they"?

Page 20, line 2: Please substitute "a nitrate increase is found" with "nitrate increases"

Page 20, line 3-4: Why is this a "plausible explaination"? Too short and speculative, this needs to be expanded and explained better, possibly in the discussion.

Page 23, line 7: Is this measurement from 1985 the only available measurement?

Page 20, line 5-15: This entire paragraph belongs to a model evaluation.

---

## Author Comment (AC1) · 29 Sep 2020

Comment:
This study investigates the effects of filaments and fronts on subduction of nitrate in the Peru-Chile Upwelling System, using a submesoscale permitting regional ocean model coupled with a biogeochemical model. The results of this study found that about 40% of the newly upwelled nitrate near the coast of Peru is subducted before being used by phytoplankton. The results show a more pronounced subduction occurs at submesoscale fronts near the cold filaments, which were underestimated by the previous mesoscale resolving models. The manuscript is well written and includes new insights on the subduction of the nitrate in EBUS. Therefore, I have only several minor comments below.

[Figure]

**Reply:**
**The authors would like to thank Takeyoshi Nagai for his overall favorable review of our manuscript and his detailed, constructive comments which helped to improve the paper. In the following we will respond individually to the detailed comments.**

Comment:
P1L11 "highlighting the additional value of direct rate measurements for model validation. " Unclear

**Reply:**
**This sentence has been rephrased and now reads as follows:**
**P1L9-11: "Surface chlorophyll in the filament is a factor 4 lower than at the upwelling front while surface primary production is a factor 2 higher."**

Comment:
P2L13 "citepChavez2007, Gruber2015, Koehn2017, Brady2019. " Typo
P11L8 "80m (100m)" Spaces are needed before "m".
P11L29 "30km" A space is needed before "km".

**Reply:**
**These mistakes have been corrected in the new manuscript.**

Comment:
P11L30 "...along-shore position matches with two SST minima (16°C) at the coast..."
I couldn't find two SST minima 16 degree at the coast in Figure 2. From the color scale
they look like 18-19 degree.

**Reply:**
**We mistakenly described the SST minima of the model simulation instead of the observations in this sentence, the temperature has been changed to reflect the observed values. It now reads as follows:**
**P16L33-34: "Their along-shore position matches with two SST minima (19°C) at the coast, suggesting that they carry recently upwelled water."**

Comment:
P14L2 ". . .is characterized in the following. . ."
"is" should be "are".

**Reply:**
**This mistake has been corrected in the new manuscript. It now reads as follows:**
**P18L4-5: "The physical structure of the filament and the distribution of the biogeochemical parameters are characterized in the following."**

Comment:
Figure 3 and 4 captions
There is no explanation for the broken white lines, which are probably MLD by reading legend, but the information should be in the captions.

**Reply:**
**The broken white lines indeed represent the mixed-layer depth as stated in the legend, the information has been added to the captions as well (Note that these figures are now Figures 4 and 5 due to the addition of Figure 2 for validation purposes following another reviewer's comment).**

Comment:
P17L28 "In brief, in situ PP observations based on carbon suggest that a significant portion of the observed offshore reduction in nitrate concentrations is due to uptake by PP. Simulations indicate that approximately 40.6% of upwelled nitrate is subducted and not utilised by PP. "
Is this because PP estimated in-situ without the effect of subduction of nitrate?

**Reply:**
**Thank you for this interesting comment. The in-situ PP estimate is computed using water samples contained in a bottle (as detailed in section 2.3). At the time and depth the bottle is closed, there is no way to determine whether the sampled water mass is a subducted water mass. To determine PP for a subducted water mass, it would be necessary to follow the water mass before it is sampled and be able to know that it has subducted. This was not done in the Eulerian experimental setup here and would require a Lagrangian approach. We can only assume that the sampled water mass corresponds to the filament and thus likely corresponds to a subducted water mass.**

Comment:
P19L32 "While the coincidence of increased subduction and a negative change in nitrate may seem counterintuitive, ..."
To me, increased subduction and reduction of nitrate are not counterintuitive.

**Reply:**
**We agree that to readers who are very familiar with the key concepts of our paper the use of the word "counterintuitive" seems out of place and is not**

**needed. We rephrased the sentence as follows:**
**P24L8-10: "The positive change in nitrate offshore together with increased eddy**
**kinetic energy (EKE) in the 1/45° simulation suggests that an increase in the**
**cross-shore nitrate flux takes over once the nitrate is removed from the surface**
**(Fig. 8a,b)."**

Comment:
P20L2 "−2 " Is the minus sign here for an increase?

**Reply:**
**The minus sign was a typo and has been fixed, the corrected sentence is:**
**P24L11-12: "At a depth of 300 m to 500 m nitrate increases near the coast (~2**
$\mu$**mol l$^{-1}$), directly below the strong negative nitrate change in the top 200 m**
**(~−2.75 $\mu$mol l$^{-1}$)."**

Comment:
P20L3 "A plausible explanation for this pattern is that organic matter is able to
sediment at shallower depth on the slope in the mesoscale simulation, which permits
enhanced local remineralization and nitrate release with respect to the submesoscale
simulation. "
I'm confused. If this is the case, remineralized nitrate is higher in 1/9 degree simulation
with negative value in Figure 7b in below 300 m near the coast?

**Reply:**
**We agree that the formulation in the first manuscript version was confusing. We**
**now describe changes in the submesoscale simulation relative to the mesoscale**
**simulation.  The key concept that was missing from our previous process**

**description is that due to increased offshore transport in the submesoscale simulation sinking organic matter sediments at deeper shelf areas, out of reach of the upwelling source waters. This results in a downward redistribution of nitrate similar to the "nutrient leakage" described in Gruber et al. 2010, a reference that we also added at this point in the manuscript. We rephrased our explanation:**

**P24L12-P25L4: "A plausible explanation for this pattern is that due to increased offshore transport in the submesoscale simulation sinking organic matter sediments on deeper shelf areas, out of reach of the upwelling source waters. This would result in a downward redistribution of nitrate similar to the "nutrient leakage" described in Gruber et al. (2011). However, small differences in the alongshore mean circulation along the continental slope (not shown) may also modify the nitrate content of the water masses. Disentangling these processes is beyond the scope of the present work."**

Comment:
P23L7 "Along-shore variability of PP in the simulations is closely related to chlorophyll concentrations, while the observed relationship of PP and chlorophyll is clearly more complex (Table 1). Investigating the reasons for this discrepancy is beyond the scope of this study and left for future work "
Is this because PP estimated in-situ without the effect of subduction to dark layers?

**Reply:**
**As explained in the reply to a previous comment, our experimental setup is not able to determine accurately whether the sampled water mass is a subducted water mass or not. So there is a possibility that the measured PP corresponds to the PP in an upwelled water mass or a subducted one. Another plausible explanation for these differences might be that despite the already high complexity in a model like PISCES, the parameterisation of primary production is**

still relatively simple. For example, changes of the phytoplankton physiology or species composition of the phytoplankton community are not modelled. Another missing process is the aging or health of the phytoplankton population, since in the real ocean phytoplankton does not change its state from "alive and containing chlorophyll" to "dead organic matter without chlorophyll" instantly. Some intermediate stage where the phytoplankton population may be hardly growing on average but still contains some chlorophyll seems more realistic, but it is not represented in this version of the PISCES model (note that a new version, PISCES-quota (Kwiatkowski et al., 2018) will be able to represent more elaborated phytoplankton stages in future studies). We have also incorporated this point into the discussion:

P28L30-P29L3: "Along-shore variability of PP in the simulation is closely related to chlorophyll concentrations, while the observed relationship of PP and chlorophyll is less clear (Table 1). Investigating the reasons for this discrepancy is beyond the scope of this study and left for future work, but we can speculate that it is related to the relatively simple parameterisation of primary production in PISCES (see section 2.5). For example, changes of the phytoplankton physiology or species composition of the phytoplankton community are not modelled. Another missing process is the aging or health of the phytoplankton population, which could slow down PP while chlorophyll concentrations remain high and thereby decouple both parameters. Note that a new version, PISCES-quota (Kwiatkowski et al., 2018) will be able to represent more elaborate phytoplankton stages in future studies."

Ref:

Kwiatkowski, L., Aumont, O., Bopp, L., & Ciais, P. (2018). The Impact of variable Phytoplankton Stoichiometry on Projections of primary production, food quality, and carbon uptake in the global ocean. Global Biogeochemical Cycles, 32(4), 516-528.

Comment:
P23L13 "Near-surface stratification in the model is by a factor of 2-3 lower than in the observations "
What mixing parameterization used in this study is missing in the manuscript. Is it KPP or two-equation type turbulence closures?

**Reply:**
**The KPP parameterization is used. We added this information in the model description section:**
**P6L29-30: "The nonlocal K-Profile Parameterization (KPP) scheme is used to handle unresolved processes related to vertical mixing."**

Comment:
P23L18-31 "However, the fact that nitrate concentration..."
I think that PISCES includes ammonium. Therefore, the authors can see the fraction of ammonium for the PP.

**Reply:**
**The PISCES model indeed includes ammonium. However, the sentence in question refers to our rate measurements which do not distinguish between new production and regenerated production. Since we have no observations to evaluate / compare the regenerated production in the model, we decided to only report total primary productivity. We rephrased the sentence slightly to make it very clear that this statement refers to observations only.**
**P28L13-18: "Besides, our observations of PP rates in the filament do not distinguish between regenerated production from ammonium and new production from nitrate (Dugdale and Goering, 1967). Fernandez et al. (2009) show that regenerated nitrogen can locally account for as much as 50% of surface primary**

**production in southern Peru. However, nitrate concentrations measured in the filament 150 km offshore are only 20 - 50% of those measured near the coast and nitrate appears to decrease with offshore distance on transect CROSS (Fig. 4). This is an indicator that a substantial amount of PP in the region of the filament survey near 15°S is fuelled by newly upwelled nitrate."**

---

## Author Comment (AC2) · 29 Sep 2020

Comment:
The authors present a study that combines observations and modeling in order to assess the role of submesoscale filaments and fronts in the lateral and vertical redistribution of the upwelled nutrients. They use ship data collected both along shore and cross shore transects off the coast of Peru to study the structure of upwelling filaments and their role in the nutrient transport and subduction, and its impact on primary production. Further, they use a coupled CROCO+PISCES simulation to strengthen these findings and expand their analysis through lagrangian experiments that allow them to follow the biogeochemical transformations of upwelled particles. They find that submesoscale filaments adevect upwelled nutrients offshore, while also

subducting them, therefore limiting production.

The paper is interesting and contains relevant in situ data on these submesoscale structures. I appreciated the Lagrangian approach to study biogeochemical transformations and the fate of the upwelled particles. The results are interesting: they strengthen previous claims regarding the importance of filaments and fronts for the offshore transport, while clarifying better the role of these structures in the enhancement of subduction (rather than submesoscale upwelling). The writing is good, with a few typos and some potential for improving the paper structure (see comments below).

I have two major comments that I'm listing below, plus a few minor/detailed comments following. My main concerns regard: lack of a thorough model evaluation, lack of a clear description of the biogeochemical model and therefore of the meaning of certain processes the authors refer to to draw conclusions. I also have a few concerns regarding the paper structure, which I am listing in the detailed comments as they refer to some specific sections. I suggest that the paper is published after revisions are done.

**Reply:**
**The authors would like to thank Referee #2 for the very thorough and fair review of our manuscript. We have taken into account the reviewer's comments and we believe this resulted in substantial improvements of our original manuscript. In particular we followed both major suggestions of the reviewer: we added a thorough model evaluation (i.e. a new figure with a Taylor diagram) and a clear description of the nitrate uptake within the biogeochemical model. All changes are outlined in detail below.**

Comment:
1) Model evaluation
In my opinion, the paper lacks a thorough model evaluation. Was this exact model setup and run used in a previous publication, where a full evaluation can be found? Is

there an appendix or supplementary material that I missed containing more information in this regard? If this is not the case, I strongly encourage the authors to provide a better evaluation of the model, as the current information doesn't seem enough to me. The only figures that provide some comparison between model and observations are Figure 2, 3, 4. However, these images refer to specific days (April 14 2017 and April 5 2017 respectively in observations and model), and they seem to show pretty large biases in the temperature and salinity fields in the model, especially in the upwelling area. These are accompanied by very high nutrient levels, suggesting a high bias in the upwelling fluxes overall. Both the physical and biochemical tracer biases can have important repercussions on the lateral and vertical fluxes and transformations which require to be discussed in the paper discussion section as they may impact the results of the Lagrangian analysis. No longer term mean evaluation of the model is explicitly provided (even though the model was run for several years). Also, the discussion of the model performance in the paper text is too qualitative and scattered throughout the text, while it would be worth having a dedicated subsection. I strongly suggest that the authors include in the paper some difference plots (SST, SSS, MLD, velocities, EKE, CHL, nutrients) and/or use a Taylor diagram to summarize the performance of the model in the region of study in order to be able to assess its limitations in the discussion.

**Reply:**
**We agree that the omission of a thorough model evaluation was an oversight on our part. Note that a thorough validation of the model has been carried out in the first author's Master's Thesis (Hauschildt 2017) which was based on the same simulations. However we agree with the review that the validation should also be included in this peer-reviewed publication. The mesoscale 1/9° model is essentially the same configuration that is used in Echevin et al. 2013, but we agree that at the very least this would need to be clarified in the text.**
**Based on your remarks we now include one figure quantifying the model fit**

**of the most relevant parameters (CHL, SST, NO3) with Taylor diagrams in the main manuscript. This is done for both the mesoscale and submesoscale model configurations. We don't want to lengthen the main paper too much and reduce the readability with the model evaluation, thus we included further information as supplementary material (Figure S1 with average horizontal maps used to compute statistics for Figure 2a,b). We now also refer to the Master Thesis for readers who are interested in even more details.**

**We also discuss in detail in several new paragraphs the impact of the subsurface nitrate bias on the amount of nitrate that is subducted (P25L21 - P27L29). We conclude that our approach tends to slightly overestimate the fraction of subducted nitrate.**

**Ref.:**

**Hauschildt, J. (2017) Observed and modeled biogeochemistry of filaments off Peru. (Master thesis), Christian-Albrechts-Universität Kiel, Kiel, Germany, 110 pp. DOI 10.13140/RG.2.2.18259.48162. http://oceanrep.geomar.de/41369"**

Comment:

2) Biogeochemical model

Throughout the text, the authors refer to key biogeochemical processes resolved by the model PISCES. However, these processes are never well defined in terms of what other modeled processes and variables they depend upon, and which quantities drive and regulate them in the model. Sometimes assumptions are made regarding modeled biogeochemical processes, when the way they work should be known from the model equations. Please, include an explicit description of the few relevant model processes in your methods section, and clarify to the reader how they relate to each other, providing fundamental knowledge regarding the characteristics of the employed bio- geochemical model. Example 1: in page 15, lines 10-16, the authors discuss the fact that chlorophyll and production relate differently to each other in model and observations; this depends on the model code, which is known, and therefore the

question is: how does PP relate to CHLA in the model? Can you explain these differences using the model equations, since they can be checked, and is it worth suggesting any amendment to the processes represented by the model given what you find in the observations? Since this process is of interest, the authors should include more information on the model representation of these processes in the methods. Example 2: page 15, lines 29-30: The authors talk about modeled primary production and nitrate uptake, however no information was provided to understand how the two relate to each other in the model. What processes contribute to nitrate uptake in PISCES?

**Reply:**
**Given that the differences in CHL and PP between simulation and observations are a major part of our discussion, we totally agree that a description of how PP and nitrate uptake are parameterised in PISCES should be part of the manuscript. We added a corresponding paragraph to the methods (section 2.5, P8L14-P9L13) which is then referenced in the discussion.**

Comment:
Page 2, line 10-11: This sentence is too long, please split it in two parts.

**Reply:**
**We split this sentence as follows and added also two more references related to organic matter remineralization and oxygen consumption off Peru:**
**P2L9-12: "Furthermore, the high productivity and export of organic matter and its subsequent remineralization at depth result in high oxygen consumption (Kalvelage et al. 2015, Loginova et al. 2018). In combination with poor ventilation by sluggish currents this leads to the presence of the shallowest and most intense oxygen minimum zone (OMZ) in the global ocean (Wyrtki, 1962; Paulmier**

**et al., 2006; Karstensen et al., 2008; Stramma et al., 2010)."**

Comment
Page 2, line 34: Please add the following modeling study for the filament transport in
the Canary Upwelling System:
Lovecchio, E., Gruber, N., & Münnich, M. (2018). Mesoscale contribution to the long-
range offshore transport of organic carbon from the Canary Upwelling System to the
open North Atlantic. Biogeosciences, 15(16), 5061- 5091.

**Reply:**
**This relevant suggested reference was added.**
**P2L32: "Previous studies have attempted to quantify the fluxes of biogeo-
chemical tracers related to eddies and filaments in EBUS using biogeochemical
models of various complexity (e.g. Nagai et al., 2015 in the California EBUS;
Frenger et al., 2018; Montes et al., 2014; Bettencourt et al., 2015; José et al.,
2017 in the PCUS; Lovecchio et al., 2018 in the Canary EBUS)."**

Comment:
Page 3, line 6-7: Please rephrase the following sentence "For instance, if nitrate uptake
by PP occurred much faster than subduction, then mainly organic matter produced
in the surface layer would be subducted, whereas if it were slower, nitrate would be
subducted."

**Reply:**
**We rephrase this sentence as follows:**
**P3L6-8: "For instance, if the time scale of nitrate uptake by PP was shorter than
that of subduction, mainly organic matter produced in the surface layer would
be subducted. If it were longer, mainly nitrate would be subducted."**

Comment:
Page 3, line 9: What does "process studies" mean?

**Reply:**
**What is typically meant by this is the following: A process study aims to increase the understanding of a particular process. In our case it is the subduction of nitrate by submesoscale turbulence. Typically process studies are carried out over shorter time periods compared to long term observations or monitoring activities which mainly aim to document the mean state and its variability. Regardless, we removed the term process study to avoid confusion:**
**P3L10-12: "Dedicated studies combining multi-disciplinary observations with modelling efforts at meso- and submesoscale are key to advance our understanding of complex physical-biogeochemical interactions (Oschlies et al., 2018)."**

Comment:
Page 3, line 23: Is 2.5 km resolution enough to resolve submesoscale? I thought one needed at least 2 km, if not 1 km. Could you please refer to literature to confirm your statement that 2.5 km is enough?

**Reply:**
**Absolute kilometer scales are not adequate to describe how much of the submesoscale spectrum is resolved. The comparison of the effective model resolution with the local Rossby Radius is more useful. Since the Peruvian upwelling system is located relatively close to the equator and the Coriolis parameter is therefore small, the Rossby radius (effectively the limit of quasi-geostrophic /**

mesoscale dynamics) is ~ 60 km (Chelton et al., 1998), an order of magnitude larger than our 2.5 km model resolution. However, we cannot claim that this model resolves all submesoscale processes. Much higher resolution on the order of 100 m would be needed to resolve e.g. symmetric instability (Thomas et al., 2008). This is precisely the reason why we call our simulation submesoscale-permitting instead of submesoscale-resolving. Only features at the upper end of the submesoscale variability spectrum such as frontal structures and filaments are resolved at this resolution, but these are the key features for subduction. We are therefore confident that our results are of interest with respect to the dynamical features and processes that we are describing. We have added a sentence relating our horizontal model resolution to the Rossby radius to the methods, section 2.5:

P7L4-7: "Since the Peruvian upwelling system is located relatively close to the equator, the Rossby radius is ~ 60 km (Chelton et al., 1998), an order of magnitude larger than our model resolution (~ 2.5 km). The Rossby radius represents effectively the limit of mesoscale dynamics and we can thus consider our model submesoscale-permitting, as it resolves the upper range of the submesoscale variability spectrum."

Ref.:

Chelton, D.B., DeSzoeke, R.A., Schlax, M.G., El Naggar, K. and Siwertz, N., 1998. Geographical variability of the first baroclinic Rossby radius of deformation. Journal of Physical Oceanography, 28(3), pp.433-460, https://doi.org/10.1175/1520-0485(1998)028<0433:GVOTFB>2.0.CO;2 Ref.:

Thomas, L. N., Tandon, A., and Mahadevan, A.: Submesoscale Processes and Dynamics, in: Ocean Modeling in an Eddying Regime, edited by Hecht, M. W. and Hasumi, H., pp. 17–38, American Geophysical Union, https://doi.org/10.1029/177GM04, 2008

Comment:

Page 3, line 26, point nr.1: As I stated, I don't think the paper answers this question, because the model evaluation in the paper is extremely limited and restricted to a few specific simulated events rather than the mean state of the system and its dynamics. Also this is more of a model evaluation problem than a scientific question. I wouldn't include question nr 1 here, I would only leave questions nr 2 and 3.

**Reply:**
**We agree and have omitted this question from the introduction.**

Comment:
Subsection 2.5: Please provide here a description of how the relevant processes discussed in the paper are represented by the biogeochemical model PISCES. Please, specify what is the time resolution of your model output.

**Reply:**
**We expanded our model description with respect to the processes governing the evolution of nitrate in PISCES (section 2.5, P8L14-P9L13). We included 5 equations: the full nitrate equation, the description of nitrate uptake by phytoplankton, the limitation terms and the full growth rate equation which controls PP. The time resolution of our model output was also added to the model description as well as the figure captions.**

Comment:
Subsection 3.1: The structure of this subsection is confusing, as it alternates results from the observations (paragraphs 1 and 3) and results from the model which actually in great part is a model evaluation. Please, move out the model evaluation from here and create a proper section that focuses on it, in order to refer to it also in the

discussion when considering the limitations of the model. Also, please list first all the results obtained from the observations, and then those obtained from the model, without switching from one to the other in each paragraph.

**Reply:**
**We changed the structure of this section so that there is no longer an alternation between observations and model results. Also, we created a new section 3.1 dedicated to evaluating the mean physical and biogeochemical fields in the model. However, we kept some description of the simulated synoptic variability in section 3.2 (section 3.1 in the previous version) in order to be compared to the synoptic observations introduced there.**

Comment:
Page 11, line 7: "The variability of biogeochemical fields is similar to observations". Please, avoid using generic terms such as "similar".

**Reply:**
**We shortened this introduction sentence and it no longer contains the unspecific term "similar". The revised sentence is:**
**P16L12: "In the simulation, the upwelling structure also dominates the variability of the biogeochemical fields (Figs. 3d; 4i,j)."**

Comment:
Page 11, line 21-24: Please, remove this summary: these results are expected in any upwelling region, so it doesn't really add much information.

**Reply:**
**We shortened the paragraph concluding this section:**

**P16L27-29: "In brief, the near-surface cross-shore gradients of temperature, nitrate and chlorophyll are well represented in the simulation. In the following section we will see how both observed and modelled cold filaments give rise to along-shore variability by advection across these gradients."**

Comment:
Figure 3 and Figure 4: What do the model output sections represent? Is this still a single instance such as Figure 2 (April 5th 2017)? Or is this some longer term mean? Please, state this in the captions.
Also, how was the specific day for the model analysis and plots chosen?
Currently, some explanation is given in page 14 lines 31- 35, however some of the plots have already been introduced before this paragraph. Also, I would assume there are several events that may resemble the observations in the many years of model run, why was only a specific event chosen rather than using an ensemble of suitable events? Please, explain your rationale for the choice of day either in the captions or in the Methods.

**Reply:**
**We agree with the reviewer that information about the time average of the model output was missing in the captions and added it. However, after careful consideration we still think that the paragraph explaining why a quantitative comparison between the model and observed filaments is difficult (lines 31-35 in the former ms) needs to be part of section 3.2 (section 3.1 in the former ms) as it leads into the synoptic comparison of observations and model. This information would be misplaced in the new model evaluation (section 3.1), since that section focuses on 2-year averaged quantities. We therefore decided to keep the paragraph at its original place but reformulated and expanded our explanation. Some information about reasoning behind the choice of model**
output has also been included in the captions of figures already introduced before this paragraph. We now also include 2-year mean along-shore averaged sections for temperature and nitrate (Fig. 2c,e) in the manuscript, which are discussed in the model evaluation section (P12L25-P13L3).

Although comparing individual synoptic events is challenging and not sufficient as mentioned correctly by the reviewer it allows us to see how these structures are represented in the model, for example parameter gradients at frontal regions etc. Thus we think it is an important part of the model evaluation and only possible if high-resolution observations as presented in this study are at hand. The changed paragraph now reads as follows:

P18L33-P19L4: "The position and shape of simulated filaments is determined largely by the mesoscale eddy field, which evolves freely in the simulation and can therefore not be expected to correspond to the variability in the real ocean at any given time. The occurrence of major upwelling events and their effect on the variability of fronts and filaments, however, is closely related to the wind forcing of the model, which was derived from satellite-based, daily ASCAT scatterometer winds. We therefore picked simulated filaments that were as close as possible in space and time to the observations, which were then taken to be representative of the area and season when the observations were carried out. The chosen filaments are comparable in scale to the observed cold filament, which had an offshore extent of 150 km - 200 km (Fig. 3a,b)."

Comment:
Page 14, line 21: The paper provides a discussion of PP and how differently it relates to CHL in model and observations. Therefore, why is PP not plotted in Fig 3 and 4 if it's a variable of interest? Would it make sense to add it?

**Reply:**

We decided to show PP in a separate table because of much fewer available observational profiles which don't lend themselves to plotting in 2D. Sampling the model output at comparable locations was more straightforward than vertical and horizontal interpolation of only a few observational data points, which would have required further assumptions. We now also decided to compute the vertical integral of the model simulations and include a comparison to satellite NPP in the model evaluation (section 3.1).

Comment:
Page 14, line 31-32: Please, rephrase the first sentence as it's not good English. Page 14, line 33: Please rephrase here with "We therefore picked simulated filaments..."

**Reply:**
**We adapted the beginning of this paragraph (P14 lines 31-35 in the previous version) according to your suggestions and moved it before the model and data are compared. It now reads as follows:**
**P18L33-P19L4: "The position and shape of simulated filaments is determined largely by the mesoscale eddy field, which evolves freely in the simulation and can therefore not be expected to correspond to the variability in the real ocean at any given time. The occurrence of major upwelling events and their effect on the variability of fronts and filaments, however, is closely related to the wind forcing of the model, which was derived from satellite-based, daily ASCAT scatterometer winds. We therefore picked simulated filaments that were as close as possible in space and time to the observations, which were then taken to be representative of the area and season when the observations were carried out. The chosen filaments are comparable in scale to the observed cold filament, which had an offshore extent of 150 km - 200 km (Fig. 3a,b)."**

Comment:
Page 15, line 10: Please, substitute similar with similarly

**Reply:**
**The sentence has been corrected:**
**P19L14-15: "PP in the modelled filament is enhanced relative to the surrounding offshore waters, similarly to observations (Table 1)."**

Comment:
Page 15, line 17-27: This paragraph is not very useful and sounds like a repetition. It could be removed.

**Reply:**
**We think it is very helpful for the reader to get a short summary of the most important points but we shortened the paragraph:**
**P19L22-26: "In summary, the simulation exhibits upwelling filaments that are similar in lateral and horizontal scale and offshore extent to those observed. Our direct rate measurements indicate that PP in the modelled filaments is comparable to observations. Despite very low chlorophyll concentrations (<0.1 mg m-$3$) in the observed filament, surface PP is by a factor 2 higher than at the upwelling front. This highlights the necessity of measuring PP in addition to chlorophyll for model validation to ensure a realistic representation of the underlying processes."**

Comment:
Page 15, line 29-30: I don't understand this first sentence. Since one can actually determine the simulated rate of nitrate uptake in the model, why should one "assume"

something about it?

**Reply:**
**You are right that nitrate uptake is available as a diagnostic from our simulation.**
**However, from the observations only total PP and nitrate concentration are**
**available thus we cannot separate nitrate uptake from ammonium uptake. We**
**commented on this aspect in the discussion (P27 lines 13-18). We rephrase this**
**statement for clarity:**
**P19L28-34: "The model is able to reproduce realistic mixed-layer nitrate concen-**
**trations as well as mixed-layer PP rates offshore in the filaments, suggesting**
**that the simulated rate of nitrate uptake (i.e. new production) is realistic as**
**well (Figs. 3e,j; 4e,j; Table 1). However, it must be noted that our PP in situ**
**estimates do not discriminate between new PP and regenerated PP resulting**
**from ammonium uptake (see discussion section)."**

Comment:
Page 17, line 14-24: This section belongs to the Methods, not the Results.

**Reply:**
**We agree with your comment and have moved this paragraph to the methods**
**(section 2.6, P10L9-19).**

Comment:
Page 17, line 24-26: Why should this implicit assumption be reasonable? Could you
please provide some reason? Also, what is the definition of "subducted" - below what
depth (this should be defined in the methods)?

**Reply:**
**A precise definition of subduction does not exist so we have to define our own. For our study, the subduction depth is defined in the methods (section 2.6, P10L14-16) as the base of the photic zone i.e. the depth of 0.1% surface PAR. Thus we consider a water parcel subducted once it leaves the photic zone. The statement about the assumption is suppressed, as it was not necessary for our calculation and confusing. We thus rephrased as follows:**
**P21L7-P22L9: "Using the subduction ratio (Equation 7) defined in subsection 2.5, we estimate that 20 days after being upwelled 40.6% of the upwelled nitrate is subducted (Fig. 6f)."**

Comment:
Page 16, line 10: This initial sentence about the importance of SST remarks that it's crucial to include a good evaluation of the temperature gradient in the model.

**Reply:**
**We agree with the comment and modified the sentence:**
**P20L11-12: "The fate of upwelled floats is closely related to their change in temperature after upwelling, thus the correct representation of the temperature gradient in the model is crucial (Fig. 3a,b)"**

Comment:
Figure 7: The x-axis labels and ticks overlap in the subplots.

**Reply:**
**The overlapping x-axis tick labels have been removed (now Fig. 8 in revised manuscript).**

Comment:
Page 19, line 1-13: Wouldn't it be necessary to also discuss how the general model performance changes changing the resolution? What about the effects linked to the change in resolution of the forcing such as the wind stress, which is known to influence both upwelling and small scale flow? This paragraph compares two runs, but none of them was thoroughly evaluated.

**Reply:**
**In response to your comments we added a separate short section for model evaluation. The wind forcing is not likely to be a source of large differences between model resolutions, since the same relatively coarse data product (AS-CAT at 1/4° resolution) is interpolated on both model grids. Only in a coupled ocean-atmosphere model simulation significant wind stress differences due to air-sea interaction at different scales would be expected. However, performing such a simulation (Ocean + Biogeochemistry + Atmosphere) at submesoscale resolution would be very expensive numerically and requires a lot of expertise for the atmospheric modelling part. This is however an interesting question that should be revisited in a future study.**

Comment:
Page19, line 20: Please add "is" in "during one season and is less..."

**Reply:**
**The sentence has been corrected, it now reads as follows:**
**P23L2-5: "Comparing the same virtual float experiment in different simulations illustrates well the relative changes in subduction and their biogeochemical**

[Figure]

impacts, but this Lagrangian approach is restricted to short integrations during one season and is less suited to obtain estimates of absolute changes of the time-averaged model fields."

Comment:
Page 19, line 31-32: Please rephrase better the sentence "Further offshore..."

**Reply:**
**This sentence has been rephrased:**
**P24L7-8: "Offshore of this negative change, an increase in nitrate ($\sim$ 1.5 $\mu$mol l-1) is apparent around 100 m depth."**

Comment:
Page 20, line 1: Who are "they"?

**Reply:**
**The formulation was not clear and we rephrased the sentence as follows:**
**P24L10-11: "The horizontal patterns of nitrate change are largest in regions of increased eddy kinetic energy (not shown).**

Comment:
Page 20, line 2: Please substitute "a nitrate increase is found" with "nitrate increases"

**Reply:**
**We followed the reviewer's suggestion:**
**P24L11-12: "At a depth of 300 m to 500 m nitrate increases near the coast ($\sim$ 2**

μ**mol l-**1**), directly below the strong nitrate decrease in the top 200 m (~ −2.75** μ**mol l-**1**)."**

Comment:
Page 20, line 3-4: Why is this a "plausible explanation"? Too short and speculative, this needs to be expanded and explained better, possibly in the discussion.

**Reply:**
**We agree with the comment. We modified this part of the text following a comment of referee 1. It is now phrased as follows:**
**P24L12-P25L4: "A plausible explanation for this pattern is that due to increased offshore transport in the submesoscale simulation sinking organic matter sediments on deeper shelf areas, out of reach of the upwelling source waters. This would result in a downward redistribution of nitrate similar to the "nutrient leakage" described in Gruber et al. (2011). However, small differences in the alongshore mean circulation along the continental slope (not shown) may also modify the nitrate content of the water masses. Disentangling these processes is beyond the scope of the present work."**

Comment:
Page 23, line 7: Is this measurement from 1985 the only available measurement?

**Reply:**
**Direct PP measurements are still fairly limited compared to concentrations of biogeochemical parameters (eg. O2, CHL). To our knowledge this is the only measurement reported in the literature that was directly comparable to our measurements due to the location, dynamical setting and high horizontal**

resolution. We now also added PP rates from Fernández et al. (2009) to the discussion, which were observed in a different season but geographically close to our study area (P27L6-10). From several other publications with PP measurements that are listed in Fernández et al. (2009), only integrated quantities are provided, which we did not calculate from our measurements. However, we calculated the vertically integrated mean PP from our simulations, to compare to the commonly used PP satellite products derived from ocean color (based on the Vertically Generalized Production Model (VGPM), Behrenfeld and Falkowski , 1997). We added some text and a reference on this in section 3.1 (P12L16-19). The corresponding sentences now read as follows:

P12L16-19: "Modelled PP in the 1/45° simulation exhibits a negative bias relative to PP estimates derived from ocean color (~ −37%). The spatial correlation of this modelled PP to the satellite estimate is high, and the standard deviation is only slightly underestimated in the simulation (r≈0.9, NSTD≈0.8), resulting in a good model fit (NRMSE≈0.45, Fig. 2a)."

P28L5-9: "The range of PP rates that were determined experimentally from incubations (Table 1) is very close to measurements by Dengler (1985) acquired during April-June 1976 in the upwelling center near 15°S (4 - 16 $\mu$mol C l-1 d-1), increasing confidence in our results. In October 2005, Fernandez et al. (2009) observed a similar range of PP rates in the top 25 m of the water column slightly north (3 - 16.5 $\mu$mol C l-1 d-1; 11 - 12°S) and south (1 - 3.5 $\mu$mol C l-1 d-1; 16 - 18°S) of our study area."

Ref.:

Behrenfeld, M. J. & Falkowski, P. G. Photosynthetic rates derived from satellite-based chlorophyll concentration. Limnol. Oceanogr. 42, 1–20 (1997). Ref.:
Fernandez, C., Farias, L. and Alcaman, M.E., 2009. Primary production and nitrogen regeneration processes in surface waters of the Peruvian upwelling system. Progress in Oceanography, 83(1-4), pp.159-168.

Comment:
Page 20, line 5-15: This entire paragraph belongs to a model evaluation.

**Reply:**
**We assume here that the reviewer refers to the paragraph on page 23, lines 5-15, not page 20. We considered moving the paragraph to the newly incorporated model evaluation, but this was not logical in the structure of the paper since the section 3.1 is placed before the primary production measurements are introduced. We therefore kept the original position but split and reformulated the paragraph as follows:**
**P28L5-9: "The range of PP rates that were determined experimentally from incubations (Table 1) is very close to measurements by Dengler (1985) acquired during April-June 1976 in the upwelling center near 15°S (4 - 16 $\mu$mol C l-1 d-1 ), increasing confidence in our results. In October 2005, Fernández et al. (2009) observed a similar range of PP rates in the top 25 m of the water column slightly north (3 - 16.5 $\mu$mol C l-1 d-1; 11 - 12°S) and south (1 - 3.5 $\mu$mol C l-1 d-1; 16 - 18°S) of our study area." P28L30-31: "Along-shore variability of PP in the simulation is closely related to chlorophyll concentrations, while the observed relationship of PP and chlorophyll is less clear (Table 1)." P29L3-9: "Vertically, the modelled PP strictly decreases with larger depth corresponding to lower light levels, while in the observations subsurface maxima of PP are found offshore and at the upwelling front (Table 1). The subsurface chlorophyll maximum is too weak and diffuse in the simulations, which is consistent with this difference in the vertical distribution of PP. This could be partly related to dynamical bias in the simulations: Near-surface stratification at the base of the mixed-layer is by a factor of 2-3 lower in the model than in the observations, which likely results in a too diffuse offshore nutricline and contributes to the weak subsurface chlorophyll maximum. For the reasons mentioned above, subsurface PP in the offshore waters is likely too low in our simulations."**

Comment:

Introduction/Discussion: Please, include the following relevant literature and discuss how it compares to your findings

FILAMENTS STRUCTURE

- Bettencourt, J. H., Rossi, V., Hernandez R Garcia, E., Marta R Almeida, M., and Lopez, C. (2017), Characterization of the structure and cross-shore transport properties of a coastal upwelling filament using three-dimensional finite-size Lyapunov exponents, J. Geophys. Res. Oceans, 122, 7433– 7448, doi:10.1002/2017JC012700.

IMPACT OF MODEL RESOLUTION ON VERTICAL FLUXES

- Zhong, Y., and Bracco, A. (2013), Submesoscale impacts on horizontal and vertical transport in the Gulf of Mexico, J. Geophys. Res. Oceans, 118, 5651– 5668, doi:10.1002/jgrc.20402.

- Zhong, Y., Bracco, A., Tian, J. et al. Observed and simulated submesoscale vertical pump of an anticyclonic eddy in the South China Sea. Sci Rep 7, 44011 (2017). https://doi.org/10.1038/srep44011

SUBDUCTION BY FRONTS

- Stukel, M.R., Song, H., Goericke, R. and Miller, A.J. (2018), The role of subduction and gravitational sinking in particle export, carbon sequestration, and the remineralization length scale in the California Current Ecosystem. Limnol. Oceanogr., 63: 363-383. doi:10.1002/lno.10636

- Michael R. Stukel, Lihini I. Aluwihare, Katherine A. Barbeau, Alexander M. Chekalyuk, Ralf Goericke, Arthur J. Miller, Mark D. Ohman, Angel Ruacho, Hajoon Song, Brandon M. Stephens, Michael R. Landry Proceedings of the National Academy of Sciences Feb 2017, 114 (6) 1252-1257; DOI: 10.1073/pnas.1609435114

**Reply:**

**We have added some of the literature you suggested to the discussion. How-**

**ever, because the discussion is already quite long and we wanted to avoid compromising the readability of our paper too much, we selected only the most relevant references.**

**P29L23-26: "More generally, Zhong and Bracco (2013) and Zhong et al. (2017) find that submesoscale frontal processes are important for the vertical tracer transports near the surface and that vertical dispersion of Lagrangian particles strongly increases with increasing horizontal model resolution, which is also in agreement with the results of this study."**

**P29L30-31: "Subduction at fronts has previously been shown in observations to be as important for organic matter export below the photic zone as gravitational sinking (Stukel et al., 2017)."**

---

## Author Response (AR2)

Report #1

Submitted on 01 Dec 2020

Anonymous Referee #2

Anonymous during peer-review:        Yes    No

Anonymous in acknowledgements of published article:    Yes    No

Recommendation to the editor

1) Scientific significance

Does the manuscript represent a substantial contribution to scientific progress within the scope of this journal (substantial new concepts, ideas, methods, or data)?

Excellent    >>Good<<  Fair    Poor

2) Scientific quality

Are the scientific approach and applied methods valid? Are the results discussed in an appropriate and balanced way (consideration of related work, including appropriate references)?

Excellent    >>Good<<  Fair    Poor

3) Presentation quality

Are the scientific results and conclusions presented in a clear, concise, and well structured way (number and quality of figures/tables, appropriate use of English language)?

Excellent    Good >>Fair<<    Poor

For final publication, the manuscript should be

>> accepted subject to minor revisions <<

Suggestions for revision or reasons for rejection (will be published if the paper is accepted for final publication)

Review of "The fate of upwelled nitrate off Peru shaped by submesoscale filaments and fronts" by Hauschildt e al.

General comment

The revised paper by Hauschildt e al. is more complete than the previous version and it is clear that the authors added quite some information to it and reorganized the results. I appreciate the new model evaluation section, and really like the new tables in the text, which help to understand the many results from the study.

However, I still find it difficult to read. The English is not polished and the new sections read badly. The structure of the paper is also odd, with the supplement consisting in one single plot without any commentary.

My suggestion is to publish the manuscript after minor revisions (especially on the writing) are done.

I will list below my comments.

**Reply: The authors would like to thank the reviewer for another thorough review of the manuscript.**

Major comments

1) English and length

- I acknowledge the overall improvement in the paper, but I still find several of its portions really difficult to read, especially the new paragraphs in bold. I really encourage all of the authors to re-read these paragraphs as well as the whole paper, and improve the quality of the writing. Otherwise, I suggest to have a mother-tongue read it and correct the writing, as this would help quite a lot to deliver the message.

I am not going to provide a full list of corrections to the writing, but I will focus on a few critical points in the detailed comments list. It is on the authors to improve the rest of the paper.

**Reply: The new revision of the manuscript has been checked again by the co-authors and some more subtle grammatical questions were corrected by a native speaker. Beyond this, we believe that the recent restructuring of problematic sections has improved the readability of the paper.**

- The manuscript is also really long, and much of the new text still includes numbers and many many details that are already provided in tables and plots. I strongly suggest that the authors make an effort to streamline and shorten the paper, where the information is already provided in plots and tables.

At the present moment, the combination of length and quality of the writing makes it very difficult to read through the entire manuscript.

**Reply: We believe that it is important to have quantitative statements directly in the text, even if this leads to duplicate information as you described. However, we improved the readability of the sections you criticized by shortening and restructuring them (e.g. reducing the number of paragraphs). The model evaluation (section 2.8) has been rewritten and the discussion (section 4) now includes subsections with titles for better readability.**

2) Model evaluation

I really appreciate that the paper now includes a model evaluation to give context to the results. This subsection may be moved to the Methods rather than the Results, so to keep the real results more in focus.

**Reply: We have moved the model evaluation from the results (section 3.1) to the methods (section 2.7).**

However, as the other "new" parts of the paper (bold parts), also this section is not quite well written and should be improved. As a general note about the content of the evaluation, the text could and should focus more on what is relevant for the present study (see comments below): it should constitute a reference to then discuss strengths and shortcomings of the model results in the discussion.
It is way more important to spend more words on these relevant aspects of the model performance, than to repeat in detail in the text all the numbers of the Taylor diagrams which already provide the information.
As the supplement consists in only one figure, I think this should rather be in an appendix to the main manuscript and not in a supplement. A supplement should be something more substantial than one spare plot.

**Reply: To our knowledge, the use of a separate supplementary file for additional figures that cannot reasonably be included in the main manuscript is fairly common in Biogeosciences. Since there are now 6 supplementary figures with several panels, including them in an appendix to the main manuscript would lengthen it by several pages. We therefore decided to keep the supplementary material separate, but added a few sentences of explanation to improve readability and provide context.**

It is good practice that, when using data (eg. AVHRR, CARS…), the authors also include the references to the relevant publications for that dataset.

**Reply: We have added the requested references at the corresponding locations.**

In terms of the content of the model evaluation:

- Please, indicate biases in absolute values rather than percentages. The plot that is currently in the supplement doesn't include difference plots, therefore this is even more important. How many degrees is the max positive SST bias, how many the negative? How many meters the MLD bias? And same for the other mentioned variables.

**Reply: We now indicate absolute values and percentages in Fig.2. We have also added difference maps for all variables in the supplementary material. The bias values are now cited in the text.**

- Given the focus of this paper is on the lateral and vertical transport, wouldn't it be appropriate to show a 2D map of the total velocity field and of the small scale variability of the flow (eg. standard deviation of SSH of the daily data or EKE)? This figure could also go in an appendix.

**Reply: We have expanded the model evaluation to include both the mean bias of EKE (P.2 L.1) and the spatial pattern of this EKE bias (Fig. S5). The patterns and bias are now discussed in the "Model evaluation" section.**

- The authors mention the upwelling being too strong in the model, and this is also visible in panels a,b,c of the supplement. This should be discussed in the model evaluation before than in the discussion.

**Reply: We have now expanded the model evaluation, by comparing model fields to observed ones. However, this type of comparison cannot be done with vertical**

**velocity, as direct observations are not available. Thus, to remain consistent with the rest of the model evaluation section, we left this paragraph in the discussion.**

- Regarding the figure in the supplement, I think comparison plots should show the difference plot (at least for the crucial 1/45o run), as without the difference the eye can easily be driven to find general similarities, but it's very difficult to see biases in a quantitative way.

**Reply: The supplementary information was expanded into multiple figures and difference maps were added for all variables.**
* * *
Detailed comments

Abstract: please add a comma after often in the second sentence "Often, studies…" or better replace with something more clear, such as "Most studies on this topic…"

**Reply: We have implemented your suggestion.**

**P1L2-4: "Most studies on this topic are either based on observations or model simulations but seldom both approaches are combined quantitatively to assess the importance of fil- aments for primary production and nutrient transport."**

Abstract: why are some sentences bold?

**Reply: We marked changes to the previous version of the manuscript in bold to help the review process, following Biogeosciences guidelines. It was not clear to us that this type of "track changes" manuscript was not welcome at this stage. In this revised version we have uploaded a regular manuscript as well as a "track changes" manuscript, as it was clearly indicated on the upload form at this stage.**

Page 2, line 5: the following sentence misses the verb in its second half "These Eastern Boundary Upwelling Systems (EBUS) are found in all major ocean basins and named after the Canary, Benguela, California, and Peru-Chile current systems."

**Reply: We now repeat the auxiliary verb "are" in the second half of the sentence for better readability.**

**P2L4-6: "These Eastern Boundary Upwelling Systems (EBUS) are found in all major ocean basins and are named after the Canary, Benguela, Cali- fornia, and Peru-Chile current systems."**

Why are some parts of the "manuscript version 3" in bold? The uploaded manuscript shouldn't be the track changes manuscript. Also, for future use, it would be nice to have the track changes manuscript at the end of the "Author's answers" file showing both what was added and what was removed from the previous submitted manuscript version. This can be obtained using latexdiff in latex, or the track changes in word. I strongly encourage the authors to use one of these methods in the future.

**Reply: The bold formatting of added text was meant to aid the review process, but we should have included and marked the deleted text as well. In general, it was not obvious to us that this type of "track changes" manuscript was not allowed or appreciated at this stage of the review process.**

Page 3, lines 6-8: The following sentence must be corrected, the English is not sound: "For instance, if the time scale of nitrate uptake by PP was shorter than that of subduction, mainly organic matter produced in the surface layer would be subducted. If it were longer, mainly nitrate would be subducted." The subordinate doesn't read correctly. One should write "… , more organic matter than nitrate would be subducted" and then "…, the opposite would be true".

**Reply: The sentence has been changed according to the suggestion.**

**P3L1-3: "For instance, if the time scale of nitrate uptake by PP was shorter than the subduction time scale, more organic matter (produced in the surface layer) than nitrate would be subducted. If it was longer, the opposite would be true."**

Page 10, lines 22-27: This paragraph is really difficult to read, please correct the English, all of the three sentences are not clear.

**Reply: We improved this section and hope it conveys its intended meaning more clearly now.**

**P10L26-29: "In this section, we verify that the two simulations realistically reproduce the annual mean physical and biogeochemical struc- tures. The model**

**evaluation focuses here on the horizontal variability over a 2-year averaging period (2015 - 2016). The com- parison of the most relevant physical and biogeochemical variables with observations is summarised in Taylor diagrams for both the 1/45◦ and 1/9◦ simulations (Fig. 2a,b). The corresponding mean horizontal fields are shown in Figs. S1-S6."**

The 2 year averaging period (last sentence here) corresponds to the analysis data, is this correct? The duration of spinup and analysis data can be mentioned in subsection 2.5 where the simulation is described, and here the authors can refer to the fact that they evaluate the mean of the 2 years of analysis data without need to mention the spinup.

**Reply: In the new manuscript only the 2-year analysis period is analysed. The spin-up is no longer mentioned in this section.**

Page 10 line 28: "The model fit of" is not a sound expression, this sentence should be rephrased.

**Reply: This section has been re-written and the sentence has been removed.**

Page 10, line 30: For me, it is not clear here what figure I should look at when the authors talk about "spatial patterns". All the discussion on patterns and biases is not clearly referring to any figure. Please, refer to the right figure when discussing the patterns. If the figure is the one in the supplement, please move it to an Appendix and mention it in the evaluation where needed.

**Reply: This section has been re-written and references to specific figures shown in the supplementary section have been added throughout.**

Subsection 3.2: This subsection sounds like another model evaluation, especially given the beginning and the new final sentence. However, as far as I understand, the authors are focusing on one particular simulated event and not an average performance of the model. This is still not clear in the text, it is only mentioned in the figure caption.

The initial sentence "The characteristic structure of coastal upwelling in the physical fields is well reproduced in our simulations" refers to figure 4, and that one single figure doesn't say much about the simulation as a whole, it only shows a particular event, therefore its description in the text shouldn't be so general.

**Reply: The sentence has been changed to reflect the fact that this statement only refers to the particular event that was compared.**

**P14L33-34: "The characteristic structure of coastal upwelling in the physical fields for this particular event is well reproduced in our simulations, but some differences exist (Figs. 4a-c,f-h)."**

At line 32, when starting the discussion of the model data, the authors should state something like (please, rephrase better before using):
From the model simulation, we chose one particular event that reproduces physical conditions similar to those of the survey and assessed the ability of the model to reproduced the dynamics observed in situ.

**Reply: We accepted the reviewers suggestion with slight changes. We would like to point out that indeed in this version of the manuscript we also compare the mean against reference datasets (section 2.8).**

**P14L32-33: "From the model simulation, we chose one particular event that reproduces physical conditions similar to those of the survey and assessed the ability of the model to capture the dynamics observed *in situ*."**

Page 25, line 4: "negative SST bias of ~ −3%", please, use absolute values in degrees. Same also a few lines after: there is a comparison between a percentage bias in nitrate and an absolute bias in previous literature. This is not good for readability. Please, include absolute values also for the biases in the present model.

**Reply: We now mention the absolute values of the SST biases. Moreover, absolute values for all biases have been added to figure 2.**

Page 25, line 30: Please, reverse the words order: "of the nitrate upwelling fluxes" or "of the upwelling fluxes of nitrate"

**Reply: The sentence has been changed as requested.**

**P27L33-34: The positive nitrate bias suggests an overestimation of the upwelling fluxes of nitrate.**